# Point-MaDi: Masked Autoencoding with Diffusion for Point Cloud Pre-training

**Xiaoyang Xiao**[1], **Runzhao Yao**[1], **Zhiqiang Tian**[2], **Shaoyi Du**[1*]

[1] State Key Laboratory of Human-Machine Hybrid Augmented Intelligence,
National Engineering Research Center for Visual Information and Applications,
and Institute of Artificial Intelligence and Robotics, Xi'an Jiaotong University
[2] School of Software Engineering, Xi'an Jiaotong University
`lopeture@stu.xjtu.edu.cn`, `rzy3320@163.com`
`zhiqiangtian@xjtu.edu.cn`, `dushaoyi@xjtu.edu.cn`

## Abstract

Self-supervised pre-training is essential for 3D point cloud representation learning, as annotating their irregular, topology-free structures is costly and labor-intensive. Masked autoencoders (MAEs) offer a promising framework but rely on explicit positional embeddings, such as patch center coordinates, which leak geometric information and limit data-driven structural learning. In this work, we propose **Point-MaDi**, a novel **Point** cloud **Ma**sked autoencoding **Di**ffusion framework for pre-training that integrates a dual-diffusion pretext task into an MAE architecture to address this issue. Specifically, we introduce a center diffusion mechanism in the encoder, noising and predicting the coordinates of both visible and masked patch centers without ground-truth positional embeddings. These predicted centers are processed using a transformer with self-attention and cross-attention to capture intra- and inter-patch relationships. In the decoder, we design a conditional patch diffusion process, guided by the encoder's latent features and predicted centers to reconstruct masked patches directly from noise. This dual-diffusion design drives comprehensive global semantic and local geometric representations during pre-training, eliminating external geometric priors. Extensive experiments on ScanObjectNN, ModelNet40, ShapeNetPart, S3DIS, and ScanNet demonstrate that Point-MaDi achieves superior performance across downstream tasks, surpassing Point-MAE by 5.50% on OBJ-BG, 5.17% on OBJ-ONLY, and 4.34% on PB-T50-RS for 3D object classification on the ScanObjectNN dataset. Codes are available at https://github.com/YangParky/Point-MaDi.

## 1 Introduction

Driven by advances in LiDAR and depth-sensing technologies, point clouds have become a fundamental data representation in applications such as autonomous driving, robotics, and virtual reality, owing to their ability to capture fine-grained geometric details of objects and environments. More recently, supervised learning has significantly advanced 3D computer vision by introducing 3D-centric methods that directly operate on raw point clouds for tasks such as object classification [34, 35, 51, 11], semantic segmentation [34, 25], and object detection [33, 26]. These approaches typically rely on large-scale annotated datasets to achieve high performance. However, unlike 2D images arranged in regular grids, point clouds lack a consistent topology, making the annotation process both expensive and labor-intensive. Labeling 3D data [4, 56, 2, 6, 60, 47] often requires expert knowledge to accurately capture complex geometrical structures, which limits the scalability and generalization ability of supervised approaches. Self-supervised learning (SSL) [29, 7, 54, 14] has emerged as a promising

---

[*]Corresponding author.

39th Conference on Neural Information Processing Systems (NeurIPS 2025).

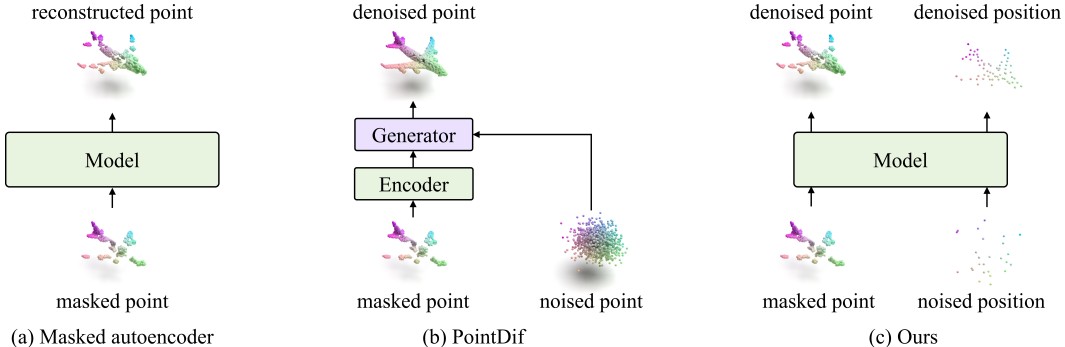

Figure 1: Comparison between different pretext tasks. (a) Masked autoencoders reconstruct masked point patches. (b) PointDif uses a conditional point generator to guide the point-to-point generation from noisy input. (c) Our Point-MaDi denoises noisy masked patches and reconstruct their centers.

alternative, enabling the extraction of generalizable representations from unlabeled point clouds through the design of various pretext tasks, including generative- and contrastive-based objectives.

Among them, diffusion probabilistic models (DPMs) [15, 28, 32] have recently emerged as a powerful paradigm for 3D point cloud representation learning, owing to their ability to model complex data distributions through iterative denoising processes. Unlike contrastive learning [57, 69, 1, 61] aligns views of a point cloud via maximizing the similarities of positive pairs to capture global semantic consistency, or reconstruction-based methods [31, 65, 9, 13, 67, 36, 37] like masked autoencoders (MAEs) [14, 31], which mask patches and reconstruct geometry using an encoder-decoder architecture. DPMs generate data step-wise from Gaussian noise and learn to reverse this process, potentially capturing rich semantic and geometric structures across scales. Despite their strengths, existing diffusion-based methods mainly rely on global context aggregation or predefined conditioning mechanisms, such as class labels or auxiliary features, to guide the denoising process. Recent studies [70, 19] have begun to address these challenges by integrating diffusion frameworks into MAEs; this structure naturally complements diffusion models: the encoder can operate on partially observed data, while the decoder can progressively recover masked content from latent noise, aligning well with the denoising objective of DPMs. Nonetheless, directly combining MAE and diffusion remains nontrivial, as current MAEs inject geometric priors, such as patch center embeddings, that leak explicit positional information into the encoder, hindering the objective of learning structure purely from data-driven cues [42, 49, 3]. This motivates our central question: **How can we design a diffusion-based pre-training framework that mitigates geometric information leakage while enhancing the modeling of local geometric structures for faithful reconstruction?**

Intuitively, if positional embeddings are particularly provided or noised, the model is forced to infer global spatial relationships solely from the visual and contextual content of the point cloud, encouraging a deeper understanding of its structure and semantics. Considering this, we propose **Point-MaDi**, a novel **Point** cloud **Ma**sked autoencoding **Di**ffusion framework. It introduces a dual-diffusion pretext task that applies diffusion to both centers and local patches, predicting centers and reconstructing patches to enforce robust modeling of comprehensive representations without external geometric cues. Fig. 1 illustrates how Point-MaDi contrasts with existing pretext tasks: while MAEs reconstruct masked patches using provided positional embeddings, PointDif advances this paradigm by formulating pre-training as a conditional point-to-point generation task. Our proposed Point-MaDi further instantiates this by performing center denoising in the encoder and patch reconstruction in the decoder, enabling geometry-aware self-supervised learning without positional cues.

To this end, we first group the point cloud into patches and apply random masking strategies to create visible and masked regions, as in traditional MAEs. In the encoder, a center diffusion process applies noise to visible and masked patch centers and tasks the model with denoising them, eliminating reliance on ground-truth positional embeddings. This process, implemented via iterative sampling, forces the encoder to model global spatial relationships by inferring center positions from partial observations. Visible patches undergo self-attention to capture intra-patch relationships, while masked patches leverage cross-attention with visible patches to model inter-patch dependencies, refining the predicted centers. In the decoder, a patch diffusion process reconstructs masked patches from latent noise, conditioned on the encoder's latent representations of visible patches and the predicted centers.

This reconstruction is optimized using Chamfer Distance, ensuring high-fidelity recovery of local structures, particularly in sparse point clouds. By integrating center diffusion for global modeling and patch diffusion for local reconstruction, Point-MaDi encourages the encoder to learn robust, context-aware representations while enabling the decoder to focus on fine-grained geometric details.

Our main contributions of this work are as follows:

- We propose Point-MaDi, a novel self-supervised pre-training framework for point clouds that disentangles positional encoding and geometry modeling via two dedicated diffusion processes, effectively mitigating positional shortcut leakage.

- The center diffusion process adds noise to both visible and masked patch centers and predicts clean coordinates, replacing explicit positional embeddings and promoting high-level spatial understanding; the patch diffusion process, conditioned on visible features and predicted centers, reconstructs noisy masked patches via step-wise denoising to recover local geometry.

- Extensive experiments on ScanObjectNN, ModelNet40, ShapeNet, S3DIS, and ScanNet demonstrate that Point-MaDi significantly outperforms existing methods in classification, segmentation, and detection tasks, validating its effectiveness and generalizability.

## 2 Related Work

**Self-supervised point cloud representation learning.** Self-supervised Learning (SSL) has achieved remarkable success in many fields such as NLP and computer vision. It aims to learn useful representations from the massive unlabeled data by defining a pre-text task [29, 17, 54] through image/patch operations, with no external labels. Recent methods have explored the potential and strengths of SSL in the point cloud domain. A prominent approach in SSL [57, 69] is contrastive learning, which encourages representations of augmented versions of an instance to be more similar compared to different inputs. PointContrast [57] pioneers contrastive learning by performing point-level invariant mapping learning. Several recent works integrate point cloud representations with other modalities, such as vision [1, 20, 46, 61] and language [66, 63, 59, 23] to facilitate the learning of transferable 3D point cloud representations. CrossPoint [1] combines intra-modal and cross-modal contrastive learning, enforcing invariance to point cloud augmentations and aligning 2D image features with point cloud prototypes. More recently, masked prediction [14] has re-attracted attention trying to recover the original input from a masked version since the introduction of the Vision Transformer (ViT) [10]. Point-Bert [62] extends the BERT-style [7] pre-training strategy to point cloud transformers via discrete Variational Autoencoder [39]. Point-MAE [31] brings the masked autoencoder idea to point cloud tasks by randomly masking input point patches and reconstructing them. Afterward, PointM2AE [65] and I2P-MAE [67] adopt hierarchical MAE transformers to extract fine-grained and higher-level semantic features of 3D shapes. Subsequent works [9] mainly enrich the model's comprehension of point cloud geometric structures using cross-modal knowledge. For instance, ACT [9] utilized a pre-trained ViT as a teacher model to acquire knowledge from other modalities. Joint-MAE [13] proposes a 2D-3D joint MAE framework to reconstruct the masked two modalities. ReCon++ [37] is similar to ReCon [36], both employing a generative framework (MAE-based) and incorporating contrastive learning through ensemble distillation. Our Point-MaDi leverages masked autoencoding with diffusion, and by progressively denoising noised positional centers, it reduces reliance on positional shortcuts while enhancing geometric awareness.

**Diffusion probabilistic models.** Diffusion probabilistic models [15, 8], also known as score-based models [43, 44], have gained significant attention in computer vision for their ability to generate high-fidelity images. DPMs begin by using an evolving Stochastic Differential Equation (SDE) to gradually add Gaussian noise to real data, turning complex data into a Gaussian distribution. Then, a time-reversed SDE maps Gaussian noise back into high-quality samples, guided by a network using the score function [45] with multiple steps. It has demonstrated superior performance in various generative fields, including image generation [28, 40, 32, 71, 18, 64], video generation [41, 12], and speech generation [21]. However, when adopted in the 3D domain, a number of works focus on 3D generation [24, 58, 27, 30, 55, 19]. Applying DPMs to 3D point cloud pre-training remains underexplored due to the challenges posed by the irregular sampling patterns of point clouds in 3D space. [24] proposed an autoencoder architecture with a DPM as a decoder. DiffPMAE [19] introduces a two-stage architecture that denoises the masked regions conditioned on the latent representation from the first stage. The most related work to ours is PointDif [70], which aggregates latent features via a condition aggregation network to guide iterative denoising of noisy point

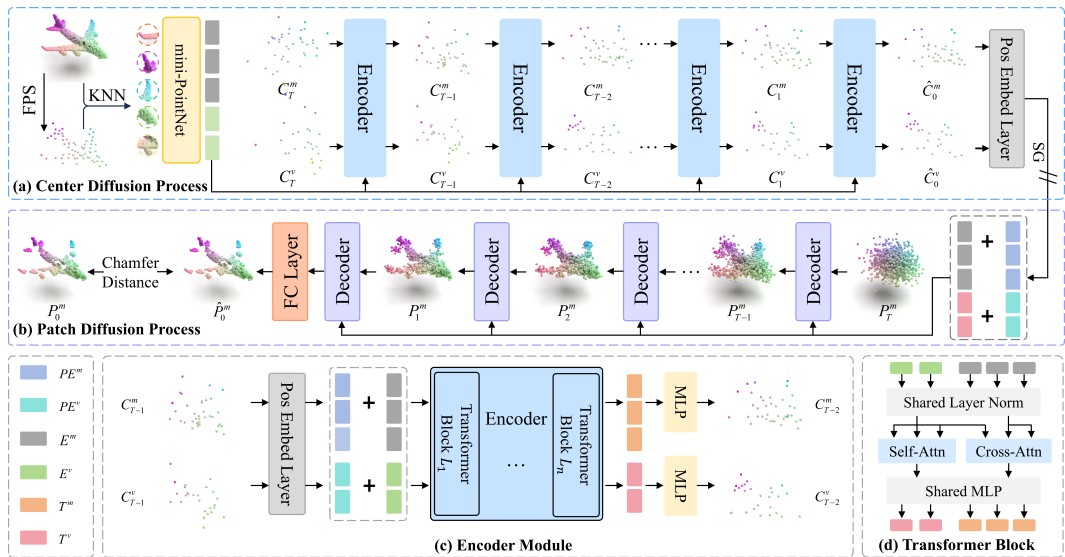

Figure 2: The pipeline of our Point-MaDi framework. The encoder adopts a center diffusion process, where noise is added to the centers of both visible and masked patches. A multilayer perceptron (MLP) maps the noisy centers to positional embeddings, and both these and visible patch embeddings are input to a transformer to predict the clean centers. The decoder performs a patch diffusion process, conditioned on the visible tokens and the predicted centers, to progressively denoise and reconstruct the masked patches. The chamfer distance guides the patch reconstruction, while the stop gradient (SG) operation prevents leakage of ground-truth masked patch positions during pre-training.

clouds for pre-training. However, our Point-MaDi differs in several key aspects. First, unlike PointDif's holistic point-to-point denoising, we introduce a joint diffusion process that simultaneously models patch centers and coordinates, capturing both structural and local geometric priors for enhanced robustness. Second, instead of relying on complex modules like PointDif's separate encoder, aggregator, and diffusion model, we unify the point encoding and decoding process with a bi-jective network design, improving efficiency and reconstruction fidelity.

## 3 Point-MaDi

### 3.1 Overview of Point-MaDi

The proposed framework contains two modules, each driven by distinct diffusion models for specific tasks. Fig. 2 illustrates the end-to-end diffusion process of Point-MaDi. For a given clean point cloud, the encoder module partitions the input into $g$ patches and trains a diffusion model that iteratively adds noise to the visible and masked patch centers. Then a denoising process predicts clean centers for all patches. The decoder module, conditioned on the encoder's visible tokens and predicted patch centers, aims to reconstruct the noisy masked patches supervised by Chamfer Distance. This dual-diffusion design enables Point-MaDi to learn both sparse structural and dense geometric representations.

### 3.2 Point cloud processing

**Point patches generation.** Following previous works [62, 31], we divide point clouds into overlapping point patches via Farthest Point Sampling (FPS) and K-Nearest Neighbors (KNN) algorithm. Formally, given the input point cloud $X \in \mathbb{R}^{n \times 3}$ with three-dimensional ($x$, $y$, $z$ coordinates) $n$ number of points, we first choose the centers $C \in \mathbb{R}^{g \times 3}$ for $g$ number of groups through FPS. For centers $C$, we adopt the KNN to select $k$ nearest points and obtain local geometric groups $P \in \mathbb{R}^{g \times k \times 3}$.

$$C = \text{FPS}(X), \quad P = \text{KNN}(X, C), \tag{1}$$

To eliminate global location bias, each patch in $P$ is normalized by subtracting its corresponding center coordinates. The resulting centered patches are treated as sub-clouds and served as a sequence of localized point representations treated like words in NLP or image patches in vision.

**Patch masking.** For the point patches, we select a predefined mask ratio $m \in (0, 1)$ and deploy random masking, outputting the visible point patches $P^v \in \mathbb{R}^{(g-r) \times k \times 3}$ and masked point patches

$P^m \in \mathbb{R}^{r \times k \times 3}$, where $r = \lfloor g \cdot m \rfloor$ is the number of masked patches, $g - r$ is the number of visible patches, and $\lfloor \cdot \rfloor$ denotes the floor function. The corresponding centers are denoted as $C^v \in \mathbb{R}^{(g-r) \times 3}$ and $C^m \in \mathbb{R}^{r \times 3}$, respectively. We also obtain a binary indicator of whether the patch is masked.

**Embedding.** The visible point patches $P^v$ are embedded through a simplified PointNet which employs 1×1 convolutions and max pooling to generate the visible tokens $E^v$.

$$E^v = \text{PointNet}(P^v), \quad E^v \in \mathbb{R}^{(g-r) \times d}. \tag{2}$$

where $d$ is the hidden dimension of the network. To incorporate spatial information, we compute positional embeddings for both visible and masked patches by applying a shared-weight MLP with GELU activation to their respective center points:

$$PE^v = \text{MLP}(C^v), \quad PE^m = \text{MLP}(C^m). \tag{3}$$

The position embeddings $PE^v$ and $PE^m$ are added to every transformer block. For the encoder, these positional embeddings are fixed, derived directly from the input patch centers to provide stable spatial cues during the masking and diffusion process. In contrast, the decoder leverages predicted center positions estimated from the encoder's output. Note that we use the same MLP for the encoder and decoder in our autoencoder to compute positional embeddings.

### 3.3 Center diffusion process

After obtaining visible and masked point patches $E^v$, $E^m$. We adopt a 12-layer standard transformer encoder as the diffusion backbone, with each block containing a multi-head self-attention (MSA) layer and a feed-forward network (FFN). The encoder applies self-attention to $E^v$ to extract conditioning features that guide the diffusion dynamics, and then performs cross-attention using $E^m$ as the query to generate masked token representations conditioned on visible context.

**Forward diffusion process.** To model geometric corruption, we apply a forward diffusion process to the centers of visible and masked point patches. At each of the $T$ time steps, Gaussian noise is incrementally added to $C^v$ and $C^m$ following a Markov chain:

$$q(C_t^v \,|\, C_{t-1}^v) = \mathcal{N}(C_t^v; \sqrt{1 - \beta_t} C_{t-1}^v, \beta_t I), \tag{4}$$

where $t_c \in [1, 2, 3, ..., T]$ is the time step of center diffusion process and $\beta_t I$ is the variance of the noise at step $t_c$, which controls the amount of noise added at each step. Since all transition kernels of the diffusion process are Gaussian, samples from the intermediate distributions can be directly formulated in a single step by applying the reparameterization trick:

$$q(C_t^v \,|\, C_0^v) = \mathcal{N}(C_t^v; \sqrt{\bar{\alpha}_t} C_0^v, (1 - \bar{\alpha}_t) I), \tag{5}$$

with $\alpha_t = 1 - \beta_t$ and $\bar{\alpha}_t = \prod_{i=1}^{T} \alpha_i$. Similarly, the forward process for masked centers can be sampled directly:

$$q(C_t^m \,|\, C_0^m) = \mathcal{N}(C_t^m; \sqrt{\bar{\alpha}_t} C_0^m, (1 - \bar{\alpha}_t) I). \tag{6}$$

We use a linear variance schedule with $\beta_t$ increasing from 0.0001 to 0.02 over 2000 steps. As time goes by, the centers gradually diffuse into a chaotic set of points. To ensure stability and consistency, we use the same variance schedule for both visible and masked centers.

**Conditional reverse process.** Providing strong conditioning information $c$ is usually helpful to reduce the number of inference steps and improve the generation quality. For visible centers and corresponding tokens, we input the latent space $E^v$ from the token layer and the corresponding noised position embedding $PE_t^v$ to generate the conditional latent visible patches.

$$T^v = \text{Encoder}(E^v, PE^v), \quad T^m = \text{Encoder}(E^m, E^v, PE^m), \tag{7}$$

where $T^v$ serves as the conditioning feature for visible centers, guiding the reverse transition, while $T^m$ serves as the conditioning feature for masked centers, integrating visible and masked patch information to guide the reverse diffusion process. It is performed via self-attention and cross-attention within each transformer block, which includes both MSA and FFN layers. The self-attention and cross-attention for a single block can be formulated as:

$$Z^v = \text{SelfAttn}(Q^v, K^v, V^v), \quad Z^m = \text{CrossAttn}(Q^m, K^{m+v}, V^{m+v}), \tag{8}$$

where $Q^v$, $K^v$, $V^v$ are the Query, Key, Value of the visible features, and $Q^m$, $K^{m+v}$, $V^{m+v}$ are the masked query, concatenated key, and value, respectively. These attention outputs $(Z^v, Z^m)$ are

passed through the FFN and normalization layers with shared parameters to produce the final encoder outputs $T^v$ and $T^m$. The self-attention and cross-attention share the parameters of transformer encoder blocks to avoid increasing parameters.

Unlike conventional diffusion models predicting additive noise, our reverse process aims to recover the visible and masked centers $C^v$ and $C^m$ by gradually removing the noise under the condition of both the visible and masked representations. However, it is non-trivial to approximate $q\big(C_{t-1}^v\big|C_t^v, T_t^v\big)$ without knowing the entire diffusion process. Therefore, we train the Encoder module with transformers to learn $p_\theta\big(C_{t-1}^v\big|C_t^v, T_t^v\big)$ for approximating the conditional probabilities to infer the entire reverse diffusion process. The reverse transition for the visible centers can be formulated as:

$$C_{t-1}^x = \left( \frac{\sqrt{\alpha_t}(1 - \bar{\alpha}_{t-1})}{1 - \bar{\alpha}_t} C_t^x + \frac{\sqrt{\bar{\alpha}_{t-1}}\beta_t}{1 - \bar{\alpha}_t} \hat{C}_0^x \right) + \sigma_t \cdot \epsilon, \quad \epsilon \sim \mathcal{N}(0, I). \tag{9}$$

Here, $x \in \{v, m\}$ indicates visible or masked centers. $\hat{C}_0^x = f_\theta(\cdot)$ denotes the predicted clean center, inferred by a shared-parameter encoder. For the visible case ($x = v$), the prediction is conditioned on $C_t^v, E^v, PE_t^v$, and $t_c$; for the masked case ($x = m$), additional inputs $E^m$ and $PE_t^m$ are included.

### 3.4 Patch diffusion process

Similar to the encoder's center diffusion process, the decoder in Point-MaDi leverages a diffusion-based approach to pre-train robust point cloud representations by denoising masked patches. However, it differs in its target and conditioning mechanism. While the encoder diffuses both visible and masked group centers ($C^v, C^m$), employing self-attention for visible centers and cross-attention for masked ones, the decoder exclusively denoises masked point cloud patches $P^m$. In the forward diffusion process, Gaussian noise is added to masked patches $P_t^m$ over $T$ time steps, formulated as $q\big(P_t^m\big|P_0^m\big) = \mathcal{N}(P_t^m; \sqrt{\bar{\alpha}_t}P_0^m, (1 - \bar{\alpha}_t)I)$, where $\bar{\alpha}_t = \prod_{i=1}^T \alpha_i$, following the same linear variance schedule as in Sec. 3.3. The reverse process approximates the denoising transition $p_\theta\big(P_{t-1}^m\big|P_t^m, T_t^v, t\big) = \mathcal{N}(P_{t-1}^m; \mu_\theta(P_t^m, T_t^v, t), \sigma_t^2 I)$, with the transformer decoder predicting clean patches. To this end, we train a diffusion model as the decoder, which conditions on the encoder's visible encoded features $T^v$ concatenated with learnable mask tokens $X^m$, utilizing a transformer-based architecture with self-attention to process these integrated inputs. This design, coupled with specialized modules for token generation and reconstruction, enables the recovery of dense local geometry, complementing the encoder's sparse center denoising, with key components as follows.

**Mask token layer.** The mask token layer maps noisy masked patches $P_t^m$ from the diffusion process to a latent representation $X_t^m \in \mathbb{R}^{r \times D}$, ensuring a constant number of output samples and aligning the masked and visible patches. It consists of a 1D convolutional layer processing input patches of shape $r \times k \times 3$. The model processes a point cloud with 1024 input points with a mask ratio $m = 0.6$, resulting in $\lfloor 1024 \cdot 0.6 \rfloor$ masked points. The output from the mask token layer is size $r \times d$.

**Time embedding.** To provide a unique embedding for each time step in the diffusion sequence, allowing the decoder transformer to learn the temporal relation and handle the time sequence. We construct a 384-dimensional frequency embedding $TE_t$ followed by a two-layer MLP with dimensionality equal to the transformer's hidden size and SiLU activations.

**Transformer decoder.** The structure of the decoder contains fewer transformer blocks compared to the encoder. Empirically, the depth of the network affects its diffusion performance. Hence, we explore fine-tuning with a different number of layers. Unlike existing SSL methods such as Point-MAE [31] and ReCon [36], which rely on ground-truth positional embeddings for both visible and masked patches alongside visible tokens ($C^v, C^m$) and learnable mask tokens $X^m$, our decoder processes a concatenated input of encoded visible tokens and noisy mask tokens, paired with positional embeddings that combine ground-truth positions for visible patches $PE^v$ and predicted positions $PE^{m,pred}$ for masked patches. This can be expressed as:

$$H^m = \text{Decoder}(T^v, X^m, PE^v, \text{SG}(PE^{m,pred}), TE), \tag{10}$$

where $H^m \in \mathbb{R}^{r \times d}$ denotes the decoder's output, SG is the stop-gradient operation. By using ground-truth $PE^v$, we leverage accurate spatial context for visible patches, while $PE^{m,pred}$, derived from the encoder's predicted centers $C^m$ via stop-gradient, prevents leakage of ground-truth masked patch positions and encourages the decoder reliance on the encoder's learned representations. The stop-gradient further ensures that decoder gradients do not disrupt the encoder's center diffusion task, preserving the encoder's robust feature representations. Subsequently, $H^m$ is passed through a

MLP layer for masked coordinate reconstruction, producing $\hat{P}^m \in \mathbb{R}^{r \times k \times 3}$. This hybrid approach enhances the robustness and generalization of patch reconstruction, complementing the encoder's sparse center denoising objective.

## 3.5 Training objective

**Center diffusion loss.** The encoder predicts clean group centers from noisy counterparts by leveraging latent features of visible and masked tokens in the forward diffusion process. It predicts visible group centers by feeding their latent features $Z^v$ into a dedicated MLP, yielding $\hat{C}^v = \mathrm{MLP}_v(Z^v)$. Similarly, it predicts masked group centers by processing $Z^m$ through a separate MLP, giving $\hat{C}^m = \mathrm{MLP}_m(Z^m)$, for visible and masked tokens, respectively. These two MLP networks do not share parameters, ensuring that each group (visible and masked) is processed independently. The loss function for this task is computed as follows:

$$\mathcal{L}_{\text{center}} = \frac{1}{g-r} \sum_{i=1}^{g-r} \left\| \hat{C}_i^v - C_i^v \right\|_2^2 + \frac{1}{r} \sum_{j=1}^{r} \left\| \hat{C}_j^m - C_j^m \right\|_2^2, \tag{11}$$

where $\hat{C}_i^v$, $\hat{C}_j^m$ and $C_i^v$, $C_j^m$ are the predicted and ground-truth visible/masked centers, respectively.

**Patch diffusion loss.** Unlike the encoder, which predicts the center positions using MSE, the decoder reconstructs masked point cloud patches from noisy inputs. The L2 Chamfer Distance is adopted as the decoder loss function.

$$\mathcal{L}_{patch} = \frac{1}{|\hat{P}^m|} \sum_{\hat{p} \in \hat{P}^m} \min_{p \in P^m} \|\hat{p} - p\|_2^2 + \frac{1}{|P^m|} \sum_{p \in P^m} \min_{\hat{p} \in \hat{P}^m} \|p - \hat{p}\|_2^2. \tag{12}$$

**Final loss.** Therefore, we have a final loss with a weighting factor:

$$\mathcal{L} = \gamma \mathcal{L}_{center} + \mathcal{L}_{patch}, \tag{13}$$

where $\gamma$ adjusts the relative importance of the center denoising task. We set it to 0.1 by default. Intuitively, the training process encourages the encoder to learn geometric features from the corrupted inputs and encourages the decoder to reconstruct the original point cloud. The diffusion process introduces structured perturbations to the data and promotes the encoder to capture both local geometric details and global context, thus enhancing its capacity beyond the original Point-MAE. After pre-training, we abandon the decoder and only keep the encoder for downstream tasks.

## 4 Experiments

### 4.1 Downstream tasks

**Linear evaluation for real-world classification.** We first fine-tune the proposed method on real-world scenes for 3D object classification. Rotation is applied for data augmentation during fine-tuning. We take the overall accuracy (OA) on ScanObjectNN [47] subsets as the evaluation metric and summarize experiment results as in Tab. 1. Our Point-MaDi achieves superior performance on all subsets, reaching 95.52%, 93.46%, and 89.52% accuracies, respectively. Compared to the previous Point-MAE [31], our diffusion-based Point-MaDi yields consistent improvements of 5.50%, 5.17%, and 4.34% on OBJ-BG, OBJ-ONLY, and PB-T50-RS, respectively. Furthermore, the performance is competitive with recent cross-modal methods (e.g., ReCon [36], I2P-MAE [67]), without requiring additional modalities or complex pre-training pipelines.

**Linear evaluation for synthetic classification.** We also conduct experiments on the classification of the synthetic ModelNet40 dataset [56]. Standard random scaling and translation are applied for data augmentation during training. While diffusion-based methods like PointDif may not consistently dominate on the relatively clean and less diverse ModelNet40 dataset, our Point-MaDi still achieves 93.8% accuracy, demonstrating strong generalization without relying on additional modalities or elaborate architectures.

**Part semantic segmentation.** We conduct part segmentation on the ShapeNetPart [60] dataset. Following previous research, we randomly sample 2,048 points from each input instance and adopt the same segmentation head for the fair comparison, which concatenates the global features with each local feature from the 4th, 8th, and 12th layers of the transformer block, and a shared MLP predicts a part label for each point. Both category mIoU and instance mIoU are computed and presented in

Table 1: Classification accuracy (%) on three variants of ScanObjectNN and ModelNet40. Parameters of inference models #P (M) are listed. We report ScanObjectNN results without voting. ModelNet40 results are shown without and with voting. Not all papers report Acc, indicated as "–" in our table.

| Method | Reference | #P | ScanObjectNN | | | ModelNet40 | |
| --- | --- | --- | --- | --- | --- | --- | --- |
| | | | OBJ-BG | OBJ-ONLY | PB-T50-RS | w/o Vote | w/ Vote |
| *Supervised Learning Only* | | | | | | | |
| PointNet [34] | CVPR 2017 | 3.5 | 73.3 | 79.2 | 68.0 | 89.2 | – |
| PointNet++ [35] | NeurIPS 2017 | 1.5 | 82.3 | 84.3 | 77.9 | 90.7 | – |
| DGCNN [51] | TOG 2019 | 1.8 | 82.8 | 86.2 | 78.1 | 92.9 | – |
| SimpleView [11] | ICML 2021 | – | – | – | 80.5±0.3 | 93.9 | – |
| PointMLP [25] | ICLR 2022 | 12.6 | – | – | 85.4±0.3 | 94.5 | – |
| P2P-HorNet [52] | NeurIPS 2022 | 195.8 | – | – | 89.3 | 94.0 | – |
| *with Single-Modal Self-supervised Learning* | | | | | | | |
| Point-BERT [62] | CVPR 2022 | 22.1 | 87.43 | 88.12 | 83.07 | 92.7 | 93.2 |
| MaskPoint [22] | ECCV 2022 | – | 89.30 | 88.10 | 84.30 | – | 93.8 |
| Point-MAE [31] | ECCV 2022 | 22.1 | 90.02 | 88.29 | 85.18 | 93.2 | 93.8 |
| Point-M2AE [65] | NeurIPS 2022 | 15.3 | 91.22 | 88.81 | 86.43 | 93.4 | 94.0 |
| PointGPT [5] | NeurIPS 2022 | 19.5 | 91.60 | 90.00 | 86.90 | – | 94.0 |
| Point-CMAE [38] | ACCV 2024 | 22.1 | 93.46 | 91.05 | 88.75 | 93.6 | – |
| PCP-MAE [68] | NeurIPS 2024 | 22.1 | 95.52 | 94.32 | 90.35 | 94.0 | 94.2 |
| PointDif [70] | CVPR 2024 | 22.3 | 91.91 | 93.29 | 87.61 | – | – |
| **Point-MaDi (Ours)** | – | 22.1 | **95.52** | 93.46 | 89.52 | 93.8 | 94.1 |
| *Improve (over Point-MAE)* | – | – | +5.50 | +5.17 | +4.34 | +0.6 | +0.3 |
| *with Cross-Modal Self-Supervised Learning* | | | | | | | |
| ACT [9] | ICLR 2023 | 22.1 | 93.29 | 91.91 | 88.21 | 93.2 | 93.7 |
| Joint-MAE [13] | IJCAI 2023 | – | 90.94 | 88.86 | 86.07 | – | 94.0 |
| I2P-MAE [67] | CVPR 2023 | 15.3 | 94.14 | 91.57 | 90.11 | 93.7 | 94.1 |
| ReCon [36] | ICML 2023 | 43.6 | 95.18 | **93.63** | **90.63** | **94.1** | **94.5** |

Table 2: Part segmentation on ShapeNetPart and semantic segmentation on S3DIS Area 5. The mean intersection over union (mIoU) for all classes (Cls.) and for all instances (Inst.) are reported for Part Segmentation. Mean accuracy (mAcc) and mIoU are reported for Semantic Segmentation.

| Method | Reference | Part Seg. | | Semantic Seg. | |
| --- | --- | --- | --- | --- | --- |
| | | Cls. mIoU | Inst. mIoU | mAcc | mIoU |
| *Supervised Learning Only* | | | | | |
| PointNet [34] | CVPR 2017 | 80.4 | 83.7 | 49.0 | 41.1 |
| DGCNN [51] | TOG 2019 | 82.3 | 85.2 | – | – |
| PointMLP [25] | ICLR 2022 | – | – | – | – |
| *Self-Supervised Representation Learning* | | | | | |
| Transformer [48] | NeurIPS 2017 | 83.4 | 84.7 | 68.6 | 60.0 |
| Point-BERT [62] | CVPR 2022 | 84.1 | 85.6 | – | – |
| MaskPoint [22] | ECCV 2022 | 84.4 | 86.0 | 70.1 | 61.0 |
| Point-MAE [31] | ECCV 2022 | 84.2 | 86.1 | 69.9 | 60.8 |
| PointGPT [5] | NeurIPS 2022 | 84.1 | 86.2 | – | – |
| PointDif [70] | CVPR 2024 | 84.4 | 85.8 | 69.5 | 60.2 |
| **Point-MaDi (Ours)** | – | **84.8** | **86.3** | **71.0** | **61.2** |
| *Improve (over Point-MAE)* | – | +0.6 | +0.2 | +1.1 | +0.4 |

Tab. 2. Our Point-MaDi achieves state-of-the-art performance, with a category mIoU of 84.8% and an instance mIoU of 86.3%, improving over Point-MAE by 0.6% and 0.2%, respectively.

**3D scene segmentation.** We validate our model on the indoor S3DIS [2] dataset to demonstrate the ability of the models to comprehend contextual semantics and intricate local geometric relationships. Tab. 2 demonstrates the performance of our proposed method. Our Point-MaDi achieves superior performance on Area 5, with a mAcc of 71.0% and a mIoU of 61.2%, improving over Point-MAE by 0.2% and 0.4%, respectively. These results underscore the effectiveness of Point-MaDi's dual-diffusion pre-training in capturing complex scene semantics and fine-grained geometric details.

Table 3: Object detection results on ScanNet. We report average precision (%). "Pre Dataset" refers to the pre-training dataset. ScanNet-Medium is a subset of ScanNet.

| Method | Reference | [P] | Pre Dataset | AP50 |
|---|---|---|---|---|
| VoteNet [33] | ICCV 2019 | × | – | 33.5 |
| STRL [16] | ICCV 2021 | ✓ | ScanNet | 38.4 |
| PointContrast [57] | ECCV 2020 | ✓ | ScanNet | 38.0 |
| 3DETR [26] | ICCV 2021 | × | – | 37.9 |
| Point-BERT [62] | CVPR 2022 | ✓ | ScanNet-Medium | 38.3 |
| Mask-Point [22] | ECCV 2022 | ✓ | ScanNet-Medium | 42.1 |
| Point-MAE [31] | ECCV 2022 | ✓ | ShapeNet | 42.8 |
| TAP [53] | ICCV 2023 | ✓ | ShapeNet | 41.4 |
| PointDif [70] | CVPR 2024 | ✓ | ShapeNet | 43.7 |
| **Point-MaDi (Ours)** | – | ✓ | ShapeNet | **43.7** |
| *Improve (over Point-MAE)* | – | - | - | +0.9 |

Table 4: Classification accuracy (%) of decoder architectures on ScanObjectNN variants. The configurations differ in how attention is applied between visible and masked tokens.

| Decoder Configuration | OBJ-BG | OBJ-ONLY | PB-T50-RS |
|---|---|---|---|
| **Joint decoder** | **95.52** | **93.46** | **89.52** |
| Cross decoder | 94.66 | 92.60 | 88.69 |
| Cross-self decoder | 93.63 | 92.43 | 87.93 |

Table 5: Classification accuracy (%) of masking strategies on ScanObjectNN variants. "Random" is random masking, "Block" is block masking, "Rand & Block" combines both.

| Masking Strategy | OBJ-BG | OBJ-ONLY | PB-T50-RS |
|---|---|---|---|
| Rand | 93.98 | 92.77 | 88.45 |
| Block | 93.63 | 91.57 | 88.10 |
| **Rand & Block** | **95.52** | **93.46** | **89.52** |

**3D object detection.** To further demonstrate the scene understanding ability of the proposed method, we fine-tune our Point-MaDi on the more challenging indoor dataset ScanNetV2 [6]. Following MaskPoint, we utilize 3DETR [26] as the baseline and replace the encoder with our Point-MaDi backbone. For evaluation purposes, we measure the Average Precision (AP) of 3D bounding boxes with 0.5 thresholds for IoU. Tab. 3 presents the results. Our method, along with Point-MAE and TAP, is pre-trained on ShapeNet in a different domain compared to the ScanNet-Medium dataset. Our Point-MaDi, pretrained on ShapeNet, achieves a state-of-the-art AP50 of 43.7%, improving over Point-MAE by 0.9%. Despite the domain gap Point-MaDi outperforms methods pretrained on ScanNet-Medium, such as MaskPoint (42.1%) and Point-BERT (38.3%), highlighting the robustness of its dual-diffusion pre-training in capturing complex scene semantics and geometric structures.

## 4.2 Ablation studies

**Decoder architecture.** We discuss the effect of different decoder designs, exploring three configurations that vary in how attention modules are applied to visible latent tokens $T^v$ and the noise tokens $X^m$. The joint decoder applies transformer blocks on the concatenated sequence of the visible latent and the noise tokens, enabling self-attention across all tokens to capture global interactions. The Cross decoder takes $T^v$ as queries and $X^m$ as keys and values in cross-attention, mapping noise tokens to reconstructed patches within visible context. The cross-self decoder combines cross-attention, where visible tokens $T^v$ serve as context, with self-attention on noise tokens $X^m$, allowing noise tokens to interact before querying visible tokens. We conduct experiments with the same depth to investigate the quality of predicted representations. As shown in Tab. 4, the joint decoder achieves the best overall performance. We make the joint decoder the default option for pre-training.

**Masking strategy.** We assess the impact of different masking strategies on the classification tasks. We trained our model with a 60% mask ratio on three masking settings: Rand: randomly selecting masked and visible parts; Block: masking a large block that contains multiple continuous and consecutive patches; and Rand & Block: which denotes that we feed both masked inputs sequentially through the same network and employ a shared weight prediction head, ensuring no additional parameters are introduced during training. As illustrated in Tab. 5, the Rand & Block strategy achieves the best performance under the same masking ratio. It introduces more spatial diversity in corrupted regions, which encourages the model to learn more robust and generalized representations.

**Effective of component.** We conduct a comprehensive ablation study focusing on the components of our dual-diffusion framework in Tab. 6. To ensure a fair comparison, we maintained the core Point-MaDi framework. The baseline uses clean patch centers for both visible and masked patches in

Table 6: Classification accuracy (%) of different component configurations on three variants of ScanObjectNN.

| Center (Vis) | Center (Mask) | Patch | Time Embedding | OBJ-BG | OBJ-ONLY | PB-T50-RS |
|---|---|---|---|---|---|---|
| - | - | - | - | 93.97 | 92.60 | 88.83 |
| ✓ | - | - | - | 93.63 | 92.43 | 88.13 |
| - | ✓ | - | - | 94.32 | 92.08 | 88.79 |
| ✓ | ✓ | - | - | 94.32 | 92.94 | 89.17 |
| - | - | ✓ | ✓ | 94.66 | 93.11 | 89.17 |
| ✓ | ✓ | ✓ | - | 94.49 | 92.43 | 88.83 |
| ✓ | ✓ | ✓ | ✓ | **95.52** | **93.46** | **89.52** |

the encoder and employs learnable mask tokens in the decoder. The key variable is the activation or deactivation of the diffusion processes. The analysis reveals several key points: firstly, the baseline achieves a performance of 93.97%, 92.60%, 88.83% on OBJ-BG, OBJ-ONLY, PB-T50-RS. When only one set of centers is noised while the other remains clean, the performance compared to the baseline is either slightly degraded or shows mixed results, suggesting that providing clean positional information for only a subset of patches creates an inconsistent learning signal for the encoder; secondly, when both visible and masked centers are noised in the encoder, performance improves to 94.32%, 92.94%, 89.17%, demonstrating that our full center diffusion mechanism, which removes all ground-truth positional shortcuts and forces the encoder to predict all clean centers from noisy inputs, is more effective than partial noising or relying on clean centers. Furthermore, this benefit is amplified when combined with the patch diffusion task in the decoder: the configuration with full center diffusion and patch diffusion enabled achieves the best overall performance, outperforming the scenario where center diffusion is disabled but patch diffusion is active. Additionally, the analysis highlights the positive impact of incorporating time embeddings for patch diffusion processes.

## 5 Conclusions

In this work, we present a novel self-supervised pre-training framework for point cloud analysis that integrates a dual-diffusion paradigm into a masked autoencoding architecture. By jointly modeling the denoising of both patch centers and masked patches, our method mitigates the risk of geometric information leakage and encourages the learning of more robust semantic and geometric representations. The proposed framework leverages a center diffusion module in the encoder to eliminate reliance on positional embeddings, while a conditional patch diffusion module in the decoder facilitates fine-grained reconstruction guided by visible context. Through extensive experiments across multiple downstream tasks, our approach consistently demonstrates superior performance and generalization. These results validate our initial motivation to force the model to infer global spatial relationships, enhancing its ability to capture comprehensive 3D structural understanding.

## 6 Acknowledgements

This work was supported by the National Natural Science Foundation of China under Grant Nos. U24A20252, and 62327808, the Key Research and Development Program of Shaanxi Province of China under Grant Nos. 2024PT-ZCK-66 and 2024CY2-GJHX-48, Guangdong Major Project of Basic and Applied Basic Research under Grant No. 2023B0303000009, and the Fundamental Research Funds for the Central Universities under Grant No. xzy022024007.

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

Table 7: Training details for pre-training and downstream fine-tuning.

| Config | ShapeNet | ScanObjectNN | ModelNet | ShapeNetPart | S3DIS |
|---|---|---|---|---|---|
| Optimizer | AdamW | AdamW | AdamW | AdamW | AdamW |
| Learning rate | 5e-4 | 2e-5 | 1e-5 | 2e-4 | 2e-4 |
| Weight decay | 5e-2 | 5e-2 | 5e-2 | 5e-2 | 5e-2 |
| Learning rate scheduler | cosine | cosine | cosine | cosine | cosine |
| Training epochs | 300 | 300 | 300 | 300 | 60 |
| Warmup epochs | 10 | 10 | 10 | 10 | 10 |
| Batch size | 128 | 32 | 32 | 16 | 32 |
| Drop path rate | 0.1 | 0.1 | 0.1 | 0.1 | 0.1 |
| Number of points | 1024 | 2048 | 1024 | 2048 | 2048 |
| Number of point patches | 64 | 128 | 64 | 128 | 128 |
| Diffusion step | 2000 | – | – | – | – |
| Beta start | 1e-4 | – | – | – | – |
| Beta end | 1e-2 | – | – | – | – |
| Augmentation | Scale&Trans+Rotation | Rotation | Scale&Trans | – | – |
| GPU device | RTX 3090 | RTX 3090 | RTX 3090 | RTX 3090 | RTX 3090 |

# Appendix

# A    Experimental Settings Details

**Pre-training details.** We sample each input 1,024 points and divide them into 64 groups, each containing 32 points. We apply scale and translation operations, followed by rotation for data augmentation. The model is pre-trained with a batch size of 128 for 300 total epochs. Following Point-BERT [62], we set the hidden dimension of each encoder block to 384, the number of heads to 6, and the FFN expansion ratio to 4. The depth of the transformer decoder is set to 4. During pre-training, we adopt the AdamW optimizer with a weight decay of 0.05 and an initial learning rate of $5 \times 10^{-4}$ with the cosine decay. All experiments are conducted on a single GeForce RTX 3090. To ensure a fair comparison, we employed identical experimental settings to the default fine-tuning. More details are provided in Tab. 7.

**Dataset details.** The ShapeNet [4] is used as our pre-training dataset; it covers over 50,000 unique 3D models from 55 common categories. ScanObjectNN [47] contains 15K unique 3D point cloud objects spanning 15 diverse categories, scanned from indoor scenes obtained by scanning, often characterized by cluttered backgrounds and occlusions. We evaluate our Point-MaDi on three variants: OBJ-BG, OBJ-ONLY, and PB-T50-RS, each with increasing complexity. ModelNet40 [56] includes 12,311 clean 3D CAD objects with 40 different categories; these objects are split into 9,843 samples in the official training set and 2,468 in the test set. ShapeNetPart [60] dataset contains 14,007 and 2,874 samples with 16 object categories and 50 semantic parts for training and validation. S3DIS [2] consists of 3D scan data from 271 rooms across 6 different indoor spaces, which are annotated into 13 classes. We evaluate our model on Area 5, while the other areas are used for fine-tuning our model.

# B    Model Efficiency Comparison

We also have conducted additional experiments comparing the pre-training efficiency of our method, Point-MaDi, against both single-modal and cross-modal masked autoencoding approaches, including the most relevant recent work PointDif. As shown in Tab. 8, we report four key metrics: the number of parameters, GFLOPs, pre-training time (hours), and downstream classification performance on ScanObjectNN and ModelNet40. Point-MaDi achieves a similar or faster pre-training time compared to PointDif (14.4h vs. 14.8h), while avoiding complex multi-stage training. Compared to other MAE-based methods, Point-MaDi introduces only a slight increase in parameters and pre-training cost, while maintaining high efficiency.

Table 8: Comparison of existing single-modal and cross-modal MAE methods in terms of pre-training efficiency and representation capability on standard SSL benchmarks.

| Method | Reference | Single/Cross Modal | Pre-training efficiency | | | Performance | |
| --- | --- | --- | --- | --- | --- | --- | --- |
| | | | # Params | GFLOPS | Time (h) | ScanObjectNN | ModelNet40 |
| Point-MAE [31] | ECCV 2022 | Single | 29.0 | **2.3** | **13.1** | 85.18 | 93.8 |
| Point-M2AE [65] | NeurIPS 2022 | Single | 15.3 | 3.7 | 29.1 | 86.43 | 94.0 |
| PointDif [70] | CVPR 2024 | Single | 25.5 | - | 14.8 | 87.61 | - |
| ACT [9] | ICLR 2023 | Cross | 135.5 | 31.0 | 52.8 | 88.21 | 93.7 |
| I2P-MAE [67] | CVPR 2023 | Cross | 74.9 | 16.8 | 64.4 | 90.11 | 94.1 |
| ReCon [36] | ICML 2023 | Cross | 140.9 | 20.9 | 28.3 | **90.63** | **94.5** |
| Point-MaDi | – | Single | 29.0 | 4.7 | 14.4 | 89.52 | 94.1 |

Table 9: Few-shot classification results on ModelNet40. We perform ten separate trials for each experimental setting and the mean accuracy (%) and standard deviation are reported.

| Method | 5-way | | 10-way | |
| --- | --- | --- | --- | --- |
| | 10-shot | 20-shot | 10-shot | 20-shot |
| *Supervised Learning Only* | | | | |
| PointNet [34] | 52.0±3.8 | 57.8±4.9 | 46.6±4.3 | 35.2±4.8 |
| DGCNN [51] | 31.6±2.8 | 40.8±4.6 | 19.9±2.1 | 16.9±1.5 |
| OcCo [50] | 90.6±2.8 | 92.5±1.9 | 82.9±1.3 | 86.5±2.2 |
| *with Single-Modal Self-Supervised Representation Learning* | | | | |
| Point-BERT [62] | 94.6±3.1 | 96.3±2.7 | 91.0±5.4 | 92.7±5.1 |
| MaskPoint [22] | 95.0±3.7 | 97.2±1.7 | 91.4±4.0 | 93.4±3.5 |
| Point-MAE [31] | 96.3±2.5 | 97.8±1.8 | 92.6±4.1 | 95.0±3.0 |
| Point-M2AE [65] | 96.8±1.8 | 98.3±1.4 | 92.3±4.5 | 95.0±3.0 |
| PointGPT [5] | 96.8±2.0 | 98.6±1.1 | 92.6±4.6 | 95.2±3.4 |
| **Point-MaDi (Ours)** | 97.2±1.9 | **99.0±0.9** | **93.5±4.3** | 95.7±2.3 |
| *Improve (over Point-MAE)* | +0.9 | +1.2 | +0.9 | +0.7 |
| *with Cross-Modal Self-Supervised Representation Learning* | | | | |
| ACT [9] | 96.8±2.3 | 98.0±1.4 | 93.3±4.0 | 95.6±2.8 |
| Joint-MAE [13] | 96.7±2.2 | 97.9±1.9 | 92.6±3.7 | 95.1±2.6 |
| I2P-MAE [67] | 97.0±1.8 | 98.3±1.3 | 92.6±5.0 | 95.5±3.0 |
| TAP [53] | **97.3±1.8** | 97.8±1.9 | 93.1±2.6 | **95.8±1.0** |
| ReCon [36] | 97.3±1.9 | 98.9±1.2 | 93.3±3.9 | 95.8±3.0 |

# C   Additional Experimental Results

## C.1   Additional downstream tasks

**3D object few-shot classification.** We conduct few-shot classification experiments on the Model-Net40 dataset to evaluate our model's ability to generalize to new categories with limited labeled data. In the $N$-way $K$-shot setting, $N$ classes are randomly selected, and $K$ instances per class are used for training. We evaluate configurations with $N \in \{5, 10\}$ and $K \in \{10, 20\}$. For each class, $K$ samples are utilized for fine-tuning the model, while 20 unseen samples per class are reserved for testing. We perform 10 independent trials for each configuration following previous works [50, 62]. The mean accuracy (%) and standard deviation across these trials are reported, as shown in Tab. 9. The results demonstrate that our method achieves superior performance compared to recent state-of-the-art approaches across various settings.

## C.2   Additional ablation studies

**Mask ratio.** We evaluate the effect of varying the mask ratio to assess its impact on downstream performance. As shown in Tab. 10, the optimum mask ratio of our Point-MaDi is 60%. In our framework, Point-MaDi learns features from the ground truth of masked patches and uses visible patches and predicted centers as guidance to reconstruct masked patches only. Hence, a lower mask ratio provides excessive visible context, which simplifies center denoising and weakens the

Table 10: Performance of our model on different masking strategies. The accuracies (%) are reported on three variants of ScanObjectNN.

| Mask Ratio | OBJ-BG | OBJ-ONLY | PB-T50-RS |
|---|---|---|---|
| 40% | 93.98 | 91.91 | 88.03 |
| 50% | 93.63 | 92.43 | 88.51 |
| 60% | **95.52** | **93.46** | **89.52** |
| 65% | 94.66 | 92.25 | 88.31 |
| 70% | 94.32 | 92.43 | 88.24 |
| 75% | 93.12 | 92.43 | 88.72 |
| 80% | 92.77 | 92.08 | 88.51 |

Table 11: Effect of different loss functions for $\mathcal{L}_{center}$ and $\mathcal{L}_{patch}$. The accuracies (%) are reported on three variants of ScanObjectNN.

| Loss Function | OBJ-BG | OBJ-ONLY | PB-T50-RS |
|---|---|---|---|
| MSE, MSE | 94.49 | 92.43 | 87.89 |
| CDL2, CDL2 | 91.91 | 90.71 | 86.47 |
| MSE, CDL2 | **95.52** | **93.46** | **89.52** |
| Smooth L1, CDL2 | 94.84 | 92.25 | 88.83 |

Table 12: The number of decoder depth. The accuracies (%) are reported on three variants of ScanObjectNN.

| Decoder Depth | # P (M) | OBJ-BG | OBJ-ONLY | PB-T50-RS |
|---|---|---|---|---|
| 1 | **24.36** | 94.49 | 91.57 | 87.75 |
| 4 | 29.68 | **95.52** | **93.46** | **89.52** |
| 8 | 36.78 | 93.98 | 90.88 | 87.54 |
| 12 | 43.87 | 93.12 | 92.60 | 88.58 |

difficulty of learning meaningful spatial correlations, while higher ratios leave too few visible patches, hindering the encoder's ability to infer accurate centers.

**Loss function.** We explore different loss function settings to evaluate the impact on the pre-training objective by varying the loss functions for $\mathcal{L}_{center}$ and $\mathcal{L}_{patch}$. We test four combinations: (1) MSE for both $\mathcal{L}_{center}$ and $\mathcal{L}_{patch}$, (2) L2 Chamfer Distance for both, (3) MSE for $\mathcal{L}_{center}$ and CDL2 for $\mathcal{L}_{patch}$, and (4) Smooth L1 for $\mathcal{L}_{center}$ and CDL2 for $\mathcal{L}_{patch}$. The results are shown in Tab. 11. The combination of MSE for $\mathcal{L}_{center}$ and CDL2 achieves the best performance across all benchmarks. This suggests that MSE effectively aligns sparse group centers to ground truth by penalizing coordinate errors directly. Using CDL2 for both yields the lowest performance. Smooth L1 for $\mathcal{L}_{center}$ mitigates outlier effects but provides less precise center predictions than MSE. Consequently, we adopt the MSE loss for center prediction and CDL2 for patch reconstruction.

**Effect of decoder depth.** We investigate the impact of decoder depth in Point-MaDi to assess whether increasing the number of transformer layers enhances the quality of patch reconstruction. The results are presented in Tab. 12. The performance improves as the depth increases from 1 to 4, indicating that a moderate number of layers helps the decoder better process the predicted representations. Nevertheless, Point-MaDi does not benefit from a larger depth, as excessive layers (8–12) lead to overfitting, focusing on overly specific patch details.

**Loss weighting.** The final loss involves a weighted combination of the center denoising loss and the patch reconstruction loss. We evaluate different weighting values to examine their influence on performance. As shown in Tab. 13, setting $\gamma$ yields the best overall results across all datasets. We also observe a consistent decline in performance as $\gamma$ increases, indicating that excessive emphasis on patch reconstruction may distract the model from learning robust and generalizable structural representations through center denoising.

**Diffusion timestep.** To measure the impact of times $t_p$ on the patch diffusion process, we pre-train the model with different values of $t_p \in \{20, 100, 200, 400, 1000, 2000\}$, keeping other hyperparameters fixed. The results are shown in Tab. 14. We observe that performance generally improves with

Table 13: Performance with different $\gamma$ values. The accuracies (%) are reported on three variants of ScanObjectNN.

| $\gamma$ | OBJ-BG | OBJ-ONLY | PB-T50-RS |
|---|---|---|---|
| 0.1 | **95.52** | **93.46** | **89.52** |
| 0.2 | 93.98 | 92.60 | 88.10 |
| 0.4 | 93.98 | 91.91 | 88.72 |
| 0.6 | 94.66 | 92.08 | 88.17 |
| 0.8 | 94.15 | 91.91 | 88.45 |

Table 14: The impact of pre-training with different timestep $t_p$. The accuracies (%) are reported on three variants of ScanObjectNN.

| $t_p$ | OBJ-BG | OBJ-ONLY | PB-T50-RS |
|---|---|---|---|
| 20 | 94.49 | 92.25 | 88.17 |
| 100 | 94.66 | 93.29 | 88.65 |
| 200 | 94.84 | 92.77 | 88.72 |
| 400 | 93.98 | 92.77 | 88.65 |
| 1000 | 93.98 | 92.94 | 88.38 |
| 2000 | **95.52** | **93.46** | **89.52** |

Table 15: The impact of timestep schedules for center $t_c$ and patch $t_p$ diffusion. The accuracies (%) are reported on three variants of ScanObjectNN.

| $t_c$ | $t_p$ | OBJ-BG | OBJ-ONLY | PB-T50-RS |
|---|---|---|---|---|
| 200 | 2000 | 94.15 | 92.60 | 88.90 |
| 1000 | 2000 | 94.15 | 92.60 | 88.72 |
| 2000 | 2000 | **95.52** | **93.46** | **89.52** |
| 200 | 1000 | 93.80 | 92.94 | 88.97 |
| 2000 | 200 | 94.66 | 91.91 | 88.72 |

increasing $t_p$, peaking at $t_p = 2000$. This trend suggests that a larger timestep introduces a broader noise range, forcing the encoder to learn robust center predictions across diverse noise levels.

**Timestep schedule.** In our default setting, center diffusion and patch diffusion are sampled simultaneously using the same number of time steps $t_c = t_p = 2000$ with a shared linear variance schedule ($\beta_t$ from 0.0001 to 0.02). We conducted ablation experiments by varying $t_c$ and $t_p$ independently. Results are shown in Tab. 15. We observe that desynchronizing the diffusion schedules leads to performance degradation in all cases. Interestingly, the model is more sensitive to changes in the decoder-side patch diffusion $t_p$ compared to variations in the encoder-side center diffusion $t_c$. When $t_p$ is reduced while keeping $t_c$ fixed (e.g., $t_p = 200$), the performance drops more significantly, likely due to insufficient corruption during training, which weakens the decoder's ability to reconstruct complex local geometry. In contrast, reducing $t_c$ while keeping $t_p$ fixed leads to a smaller, though still noticeable, performance drop.

**Time embedding of the encoder.** In our framework, we intentionally omit time-step embeddings in the transformer encoder during the center diffusion process. The motivation behind this decision is to align the encoder's architecture with downstream tasks, where no diffusion steps or time conditioning are present. By avoiding the introduction of time embeddings during pre-training, we ensure that the encoder learns time-agnostic, task-agnostic features that generalize better across downstream domains. To validate this design decision, we conducted an ablation study comparing two variants in Tab. 16: 1) With time embedding added to the encoder; 2) Without time embedding, as used in our default implementation. The results on the ScanObjectNN benchmark are summarized below. Incorporating time embeddings led to consistent performance degradation across all three variants. We hypothesize that the time embeddings may introduce unnecessary conditioning noise or reduce the encoder's ability to generalize to downstream inputs, which are always clean and have no associated time-step semantics.

Table 16: The effect of time embedding in the encoder. The accuracies (%) are reported on three variants of ScanObjectNN.

| Time Embedding | OBJ-BG | OBJ-ONLY | PB-T50-RS |
|---|---|---|---|
| × | 94.49 | 92.59 | 88.83 |
| ✓ | **95.52** | **93.46** | **89.52** |

## D  Additional Visualization Results

We present additional visualizations on the ShapeNet [4] test split to evaluate the denoising performance of our Point-MaDi model. The encoder independently predicts visible and masked center points from 3D input coordinates, which are concatenated to form the denoised centers. Concurrently, the decoder reconstructs the full point cloud from encoded features. As illustrated in Figures 3–5, the denoised centers closely align with the GT centers across diverse categories, demonstrating the encoder's ability to capture global structure effectively. Meanwhile, the decoder reconstructs both global shape and local geometry in the denoised points. These results highlight the model's strengths in global representation and reconstruction, with ongoing challenges in detailed local recovery.

## E  Limitations

This work primarily focuses on learning robust geometric and semantic representations from point clouds in a purely self-supervised and unimodal setting. While the proposed dual-diffusion framework achieves consistent improvements across various downstream tasks, several directions remain open for future exploration. One natural extension is to incorporate multi-modal information, such as images or language, to further enrich the learned representations and enhance scene understanding. Additionally, while our encoder effectively predicts denoised centers that capture the global shape of point clouds, it struggles to preserve fine-grained local details, limiting the precision of local geometry in the predicted centers.

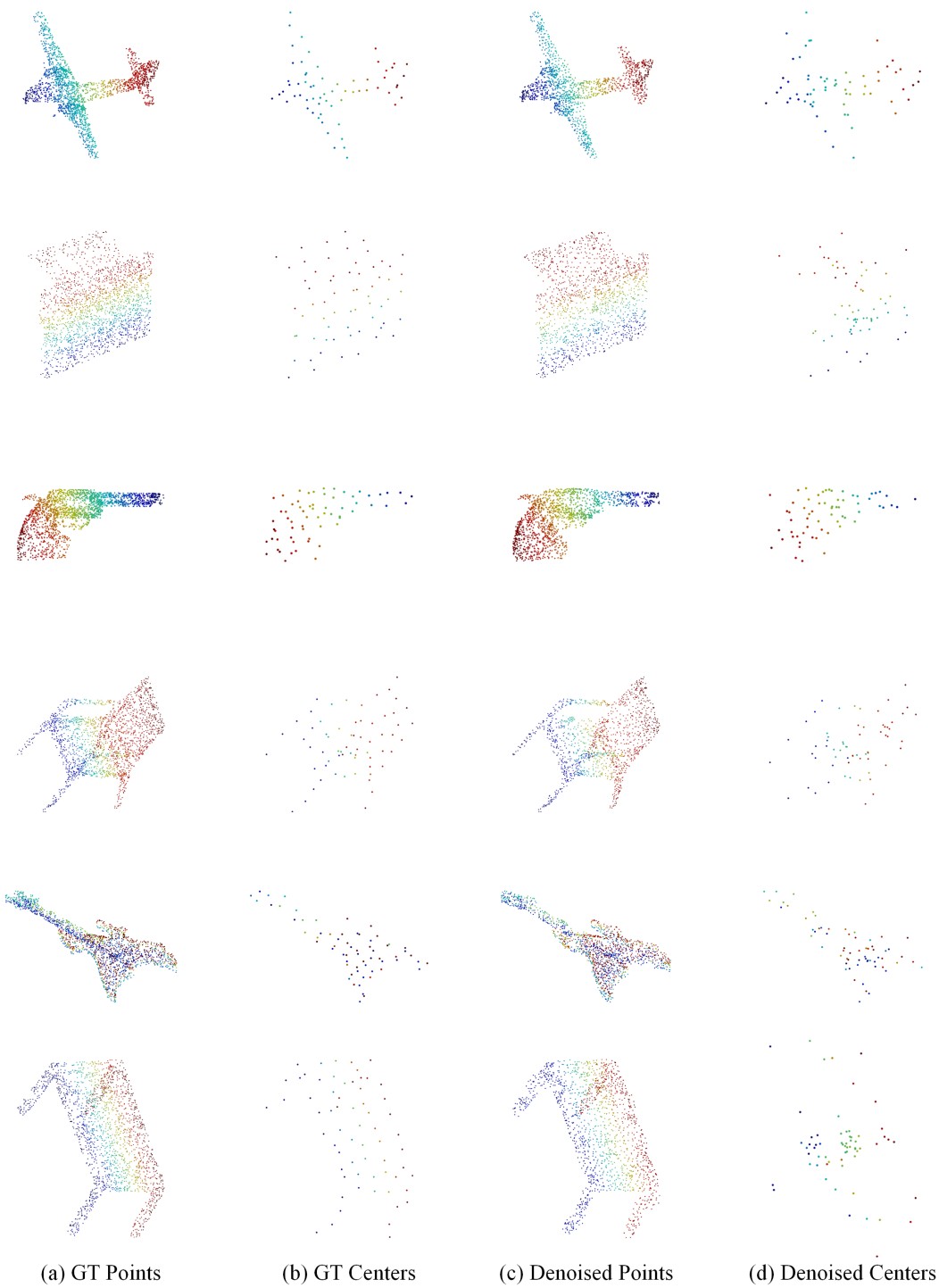

(a) GT Points          (b) GT Centers          (c) Denoised Points          (d) Denoised Centers

Figure 3: Visualization of point cloud denoising by Point-MaDi. (a) GT Points: original point cloud on ShapeNet test split. (b) GT Centers: FPS-sampled centers. (c) Denoised Points: decoder-reconstructed point cloud. (d) Denoised Centers: encoder-predicted center points.

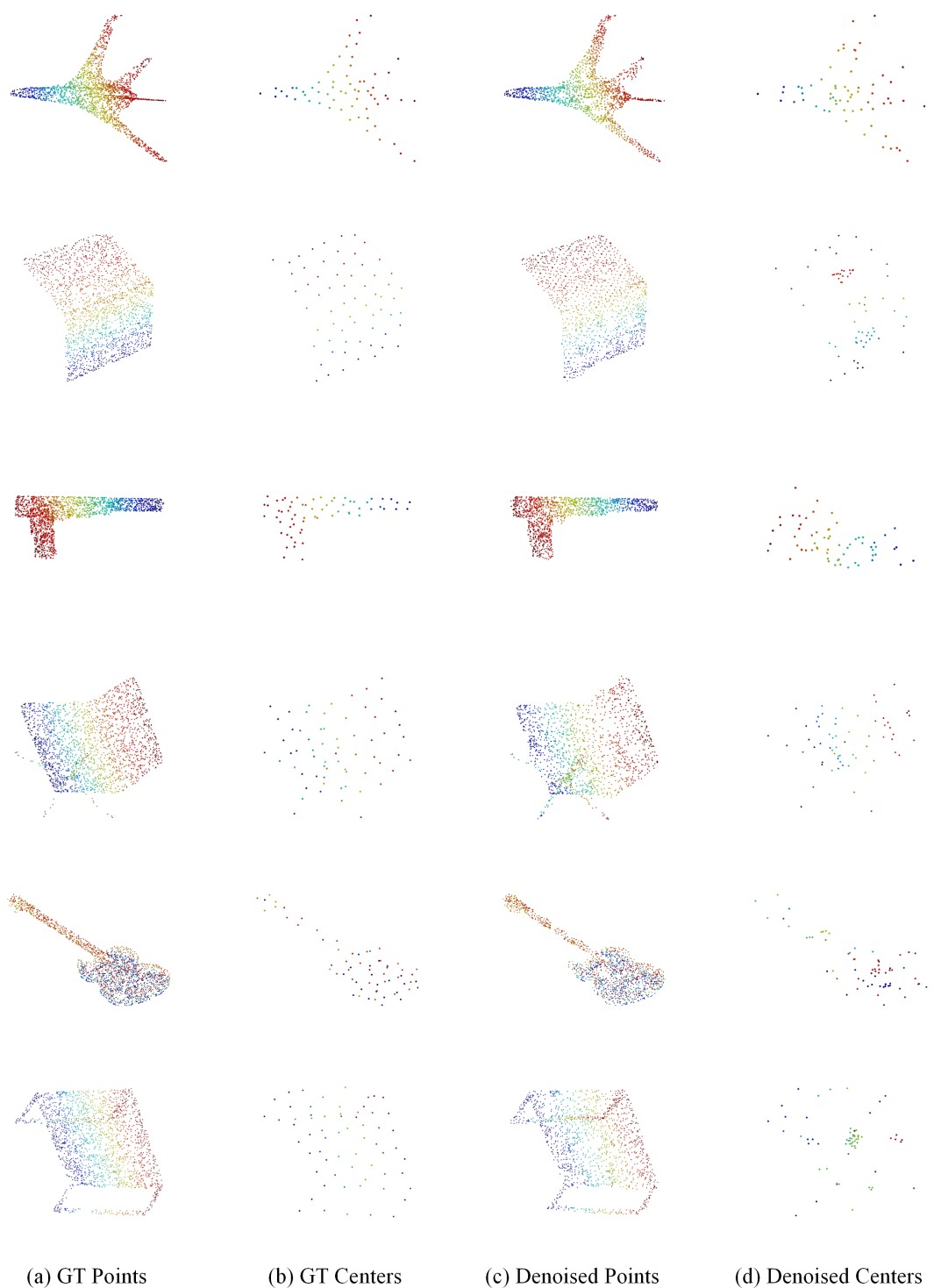

(a) GT Points      (b) GT Centers      (c) Denoised Points      (d) Denoised Centers

Figure 4: Visualization of point cloud denoising by Point-MaDi. (a) GT Points: original point cloud on ShapeNet test split. (b) GT Centers: FPS-sampled centers. (c) Denoised Points: decoder-reconstructed point cloud. (d) Denoised Centers: encoder-predicted center points.

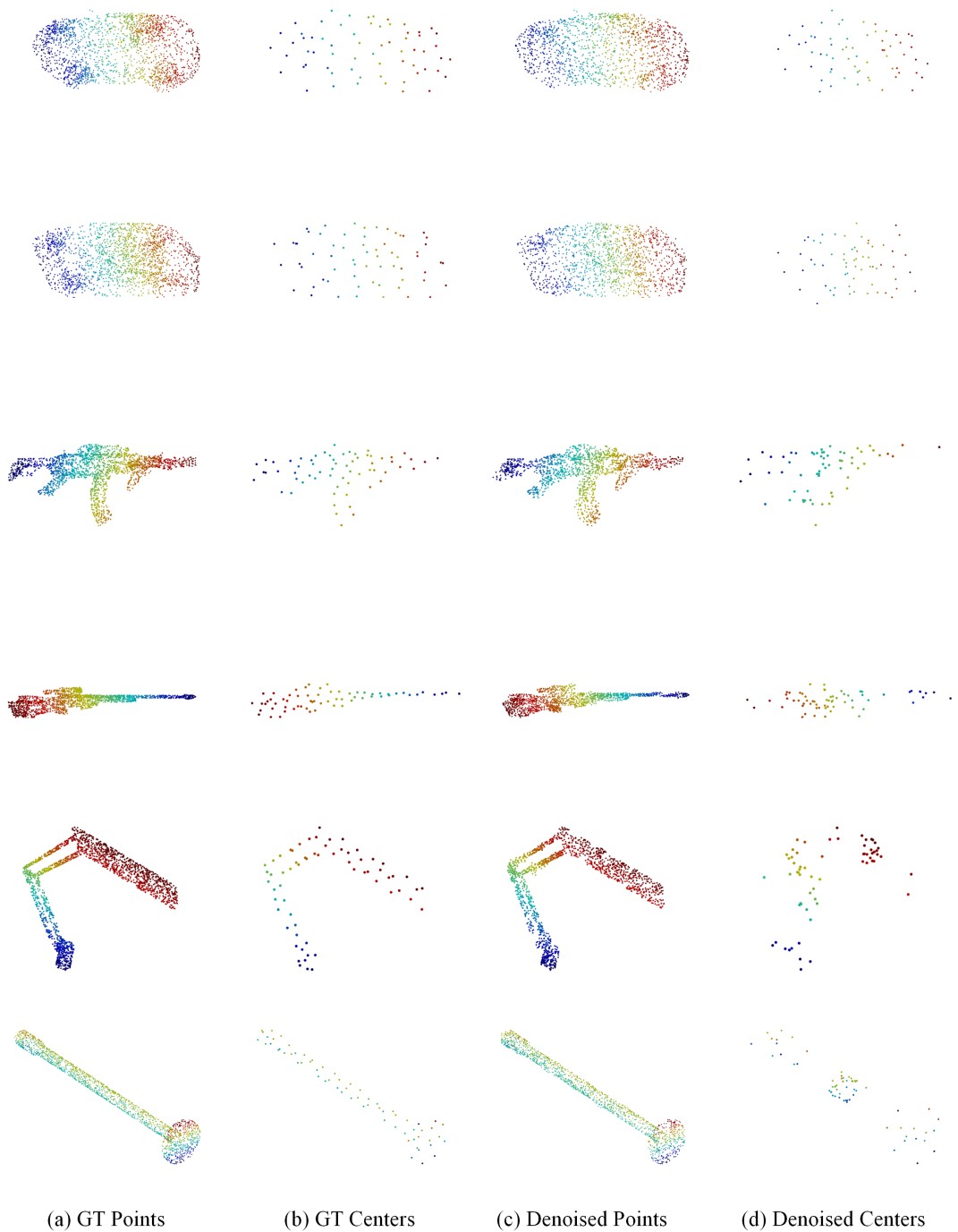

    (a) GT Points          (b) GT Centers        (c) Denoised Points    (d) Denoised Centers

Figure 5: Visualization of point cloud denoising by Point-MaDi. (a) GT Points: original point cloud on ShapeNet test split. (b) GT Centers: FPS-sampled centers. (c) Denoised Points: decoder-reconstructed point cloud. (d) Denoised Centers: encoder-predicted center points.

