# OpenReview forum: "Point-MaDi: Masked Autoencoding with Diffusion for Point Cloud Pre-training"
_NeurIPS.cc/2025/Conference — NeurIPS 2025 poster_

### Official Review · Reviewer_LUo6 · 2025-06-27

**Clarity:** 4
**Significance:** 4
**Originality:** 4
**Rating:** 5
**Confidence:** 3

**Summary:**

The proposed Point-MaDi is a novel masked autoencoding diffusion framework for self-supervised 3D point cloud representation learning, which introduces a dual-diffusion mechanism to eliminate reliance on explicit positional embeddings. Its center diffusion encoder predicts patch coordinates without ground-truth geometric priors, while the conditional patch diffusion decoder reconstructs masked patches directly from noise using latent features, enabling pure data-driven learning of both global semantics and local geometry. Experiments show state-of-the-art performance and demonstrate superior generalization across multiple 3D tasks.

**Questions:**

In addition to the ones mentioned in weaknesses, please solve the following confusions:
1. In the visualization in Appendix C, it seems that the reconstruction of the center point of some samples is relatively poor. Can authors provide the performance of center point prediction of different categories, such as CD distance, etc., and put it together with the accuracy of different categories to conduct a more in-depth analysis of the center diffusion process?

**Ethical Concerns:**

["NO or VERY MINOR ethics concerns only"]

**Limitations:**

yes

**Quality:**

4

**Strengths And Weaknesses:**

**Strengths**:
1. The proposed Center Diffusion avoids using real position embedding by predicting the coordinates of image blocks, effectively preventing the leakage of geometric information, thereby improving the model's ability to autonomously locate local areas.
2. This method uses latent features to reconstruct obscured image blocks from noise, enhancing the model's understanding and generation of local structures, enabling it to more accurately restore missing content.
3. This method abandons external geometric priors and relies only on the data itself for learning, thereby optimizing both global and local representations and improving the generalization and adaptability of the model.

**Weaknesses**:
1. Lack of model efficiency comparison. In the main experimental results table 1, please provide a comparative analysis of different methods in terms of computational complexity, and inference speed, so as to have a more comprehensive understanding of model performance.
2. Lack of visual analysis of segmentation tasks. Lack of visual analysis on segmentation tasks, as well as visual comparative analysis with other methods.

---

> ### Author Rebuttal · Authors · 2025-07-29
>
> We sincerely thank the reviewer for the helpful suggestions.  Below, we provide point-by-point responses to the raised concerns and describe the corresponding improvements made in the revised manuscript.
>
> **\<W1\> Lack of model efficiency comparison.**
>
> We sincerely thank the reviewer for the insightful suggestion regarding model efficiency. To provide a more comprehensive understanding of Point-MaDi’s performance, we have conducted additional experiments comparing the pre-training efficiency of our method, Point-MaDi, against both single-modal and cross-modal masked autoencoding approaches, including the most relevant recent work Point-Diff. We report four key metrics: the number of parameters, GFLOPs, pre-training time (hours), and downstream classification performance on ScanObjectNN and ModelNet40. Point-MaDi achieves a similar or faster pre-training time compared to Point-Diff (14.4h vs. 14.8h). Compared to other MAE-based methods, Point-MaDi introduces only a slight increase in parameters and pre-training cost, while maintaining high efficiency.
>
> | Method         | Reference    | Modal Type  | # Params | GFLOPs  | Time (h) | ScanObjectNN | ModelNet40 |
> | -------------- | ------------ | ----------- | -------------------- | ------- | -------- | ------------ | ---------- |
> | Point-MAE      | ECCV 2022    | Single      | 29.0       | 2.3   | 13.1    | 85.18        | 93.8       |
> | Point-M2AE     | NeurIPS 2022 | Single      | 15.3                 | 3.7     | 29.1       | 86.43        | 94.0       |
> | Point-Diff     | CVPR 2024    | Single      | 25.5                 | 4.7       | 14.8       | 87.61        | –          |
> | ACT            | ICLR 2023    | Cross| 135.5                | 31.0    | 52.8       | 88.21        | 93.7       |
> | I2P-MAE        | CVPR 2023    | Cross| 74.9                 | 16.8    | 64.4       | 90.11        | 94.1       |
> | ReCon          | ICML 2023    | Cross| 140.9                | 20.9    | 28.3       | 90.63        | 94.5       |
> | Point-MaDi | Ours     | Single  | 29.5             | 5.4 | 14.4   | 89.52    | 94.1   |
>
>
> **\<W2\> Lack of visual analysis of segmentation tasks.**
>
> We thank the reviewer for the constructive suggestion regarding visual analysis on segmentation tasks. We acknowledge that the current version lacks qualitative comparisons in this aspect, and we agree that such visualizations would provide valuable insight into the effectiveness of the learned representations. Due to the restrictions on adding figures during the rebuttal phase, we are unable to include segmentation visualizations at this time. However, we will provide detailed qualitative comparisons on the S3DIS dataset in the appendix of the final version. These results will highlight the spatial coherence and boundary precision of our segmentation outputs, as well as visual differences compared to existing baseline methods.
>
> **\<Q1\> Can authors provide the performance of center point prediction of different categories？**
>
> We thank the reviewer for the careful observation. We agree that some samples in Appendix exhibit relatively poor center reconstruction quality, and we appreciate the suggestion to conduct a more fine-grained analysis based on object categories.
>
> We computed the per-category Chamfer Distance * $10^2$ (CD) over a subset of representative categories from the ShapeNet test set. The results reveal that categories with clearer and more rigid structures, such as guitar, rifle, and airplane, achieve significantly better reconstruction accuracy. In contrast, categories with less consistent shapes or small-scale geometry, such as earphone and table, show larger prediction errors. These findings align well with the visual examples in Appendix, where certain object types (e.g., earphones or tables) exhibit noisier center outputs. This analysis supports our interpretation that the center diffusion process captures global geometric priors, but may be challenged by fine-grained or ambiguous shapes. We will include this category-level CD analysis in the final version to complement the visual results and provide further insight into the behavior of center diffusion across different object types.
>
> | Category     | microphone | earphone | guitar | laptop | table | chair | airplane | car  | rifle | motorbike | pistol |
> | :----------- | :--------: | :------: | :----: | :----: | :---: | :---: | :------: | :--: | :---: | :-------: | :----: |
> | CD ↓         |    2.60    |   4.78   |  0.73  |  2.83  | 4.07  | 2.67  |   1.43   | 2.29 | 1.04  |   2.68    | 0.02 |

---

### Official Review · Reviewer_wMGd · 2025-06-28

**Clarity:** 3
**Significance:** 3
**Originality:** 3
**Rating:** 5
**Confidence:** 5

**Summary:**

In this paper, the authors propose a point cloud pre-training method that combines masked auto-encoding and diffusion. They claim that this can help to avoid geometric information leakage in point-MAE-style pre-training. They conduct extensive experiments and ablations to verify the effectiveness of the pre-training method.

**Questions:**

Question 2 is the most crucial one and I'm expecting the authors' rebuttal.

1. What is the implementation detail of the PointNet embedding process of visible point patches? Do you strictly avoid masked information leakage in this encoding process?

2. Since the positional embedding of masked patches is used as conditional latent to the denoising decoder, I think the masked center information is also leaked in the reconstruction process. Is there any misunderstanding? From my perspective, if you want to claim that geometric information leakage is avoided, then no masked center features should be provided. The diffusion model should learn how to denoise masked regions only based on visible patches. If not, I think this paper would be overclaimed.

3. Why can you use MSE loss instead of CHamfer distance on center diffusion loss? Is the center point order kept during the diffusion process?

4. More recent related works should be included in the results comparison: Point-CMAE[1], PCP-MAE[2].

5. Whether this method is applicable to scene-level model pre-training? For example, directly pre-training on ScanNet dataset like PiMAE[3]?

#### References:

[1] Bin Ren, Guofeng Mei, Danda Pani Paudel, Weijie Wang, Yawei Li, Mengyuan Liu, Rita Cucchiara, Luc Van Gool, and Nicu Sebe. Bringing masked autoencoders explicit contrastive properties for point cloud self-supervised learning. arXiv preprint arXiv:2407.05862, 2024.

[2] Xiangdong Zhang, Shaofeng Zhang, and Junchi Yan. Pcp-mae: Learning to predict centers for point masked autoencoders. arXiv preprint arXiv:2408.08753, 2024.

[3] Chen, Anthony, et al. Pimae: Point cloud and image interactive masked autoencoders for 3d object detection. Proceedings of the IEEE/CVF Conference on Computer Vision and Pattern Recognition. 2023.

**Ethical Concerns:**

["NO or VERY MINOR ethics concerns only"]

**Final Justification:**

The authors have addressed my concerns in the rebuttal, particularly regarding their explicit approach to preventing information leakage. As this is an important issue for the community, I’m glad to see the paper tackling it explicitly. I therefore recommend acceptance of this paper.

**Limitations:**

Yes.

**Paper Formatting Concerns:**

N/A.

**Quality:**

3

**Strengths And Weaknesses:**

#### Strengths:

1. The motivation to resolve the geometric information leakage problem is crucial to the point cloud community.

2. The paper is overall well-written and easy to follow.

3. The experiment results convince the effectiveness of the proposed method.

#### Weaknesses:

1. The technical contribution of this paper is limited. It simply combines point-MAE with the diffusion model.

2. Whether the geometric information leakage problem is resolved is questionable.

---

> ### Author Rebuttal · Authors · 2025-07-30
>
> We deeply appreciate the reviewer’s insightful questions and in-depth observations. The issues raised prompted a deeper reflection on the core design of our method and offered us an opportunity to clarify important aspects. We provide comprehensive responses and supporting evidence to each point below.
>
> **\<Q1\>** We provide the implementation details and clarify the design rationale.
>
> 1). Our embedding module adopts a standard PointNet-like architecture, following previous works on patch-level point cloud encoding. The encoder consists of two sequential 1×1 convolutional blocks with BatchNorm and ReLU activations, followed by global max pooling to produce fixed-size per-patch representations. Each patch ${{P}^{v}}$, ${{P}^{m}}$ is encoded independently using only its own points, with no cross-patch aggregation.
>
> 2). While we do encode both visible and masked patches in the encoder, we emphasize that this design is not a source of masked information leakage, but rather a preventive mechanism that enables spatially informed yet leakage-free learning. This choice is motivated by three key aspects: (1) The center denoising task is a self-supervised pretext objective, analogous to masked language modeling or masked image modeling, where masked inputs are still processed during learning. (2) The decoder, which performs reconstruction, is the module where leakage must be strictly avoided. We ensure that the decoder has no access to masked patch features or ground-truth positions. Positional embeddings are derived from predicted centers with stop-gradient applied, ensuring that gradients cannot flow back to the encoder. (3) During encoder training, gradients from center denoising do backpropagate through masked patch features, but this occurs only in the context of pretext learning, not reconstruction. The model is not using masked features to predict visible outputs or reconstruct their geometry.
>
> An additional question might be whether it is feasible to predict all centers using only visible patch tokens. While this may work for visible centers, it becomes problematic for masked ones due to two main reasons: (1) Masked patches lack corresponding tokens, making it difficult to associate visible features with spatially distant masked centers; (2) The number and identity of masked centers do not align with visible tokens, introducing ambiguity and instability in training targets.
>
> **\<Q2\>** We thank the reviewer for raising this important and nuanced concern regarding potential geometric information leakage and the proper conditioning in our masked patch denoising process, and we appreciate the opportunity to clarify our design.
>
> **1). Regarding the concern of geometric information leakage in our encoder-decoder architecture.** In our design, we explicitly prevent geometric leakage of masked centers by the following strategies:
>
> In the encoder, we apply a center diffusion process that perturbs both visible and masked patch centers via Gaussian noise. The encoder is then trained to denoise both sets of centers. This dual prediction setup ensures that center positions are approximated, not copied, thereby preserving the self-supervised nature of the task. In the decoder, positional embeddings for masked patches are derived from the predicted centers, and a stop-gradient operation is applied. This ensures that the encoder's predicted centers do not receive gradient signals from the decoder, further preventing any potential shortcut or information leakage via backpropagation.
>
> Thus, while both visible and masked patch features are used to predict masked centers, these centers are never used in their ground-truth form during reconstruction, and no leakage path exists between the input and output. Our design differs from traditional MAE approaches where ground-truth patch center coordinates are directly provided as positional embeddings to the encoder or decoder.
>
> **2). Regarding whether masked regions are denoised solely based on visible patches.** We affirm that our decoder strictly follows this principle. During decoding, the reconstruction of masked patches is conditioned only on: (1) The visible latent features (Tᵛ) produced by the encoder; (2) The masked tokens, which are initialized as Gaussian noise and progressively denoised; (3) The positional embeddings derived from the encoder’s predicted centers, with stop-gradient applied.
>
> **3). Regarding the use of positional information in the decoder and overclaiming concerns.** We totally understand the reviewer’s concern that the presence of positional information in the decoder may seem to contradict our claim of avoiding geometric priors. However, we respectfully clarify that:
>
> Our method does not eliminate positional embeddings altogether; rather, we avoid explicit ground-truth positional shortcuts. In the decoder, we employ positional embeddings derived from predicted centers, not ground-truth ones. These predicted positions are learned through the encoder’s center diffusion process and are not tied to any external priors. Furthermore, using predicted positional embeddings is a necessary design choice, both theoretically and empirically. Transformer architectures lack built-in inductive bias for spatial structure. Without position encodings, tokens become permutation-invariant, and the network is unable to distinguish between geometrically distinct patches with similar appearance. This makes reconstruction ill-posed and ambiguous, especially in 3D point clouds where spatial relations are critical.
>
> More importantly, in our framework, the model is tasked with learning two intertwined objectives during pre-training: (1) Predict the center positions. (2) Reconstruct the geometric structure of those masked patches. These are mutually dependent tasks. If the model is denied both ground-truth and predicted positional cues, it must simultaneously infer what to reconstruct and where, without any explicit spatial reference. This constitutes a significantly under-constrained problem, leading to poor convergence and inferior downstream transferability. Additionally, excluding all positional embeddings introduces a misalignment between pre-training and fine-tuning: downstream tasks such as classification or segmentation inevitably rely on positional context, and models pretrained without any form of positional grounding tend to generalize poorly.
>
> Therefore, we believe that the use of predicted, detached positional embeddings strikes a principled balance between geometric integrity and task feasibility, and should not be interpreted as an overclaim.
>
> **\<Q3\>** We provide a detailed response to each part of this question as follows:
>
> **1). Regarding using MSE loss instead of Chamfer distance.** We adopt MSE loss for the center diffusion task due to the structured nature of the prediction targets. The center diffusion process aims to predict the clean coordinates of patch centers from their noisy counterparts. These centers, obtained via FPS as described in Section 3.1, have a fixed and unique correspondence to specific patch indices in the point cloud. Unlike the patch diffusion task, which reconstructs entire point patches and must account for unordered point sets, the center diffusion task operates on a structured set of 3D coordinates with a one-to-one mapping to the ground-truth centers. MSE is well-suited for this task because it directly measures the Euclidean distance between predicted and ground-truth center coordinates, ensuring precise alignment without requiring permutation invariance, which Chamfer Distance is designed to handle. As shown in Table 6, using MSE for the center diffusion loss outperforms Chamfer Distance, achieving higher accuracies across all ScanObjectNN variants. This suggests that MSE effectively encourages the encoder to learn robust spatial relationships for center prediction.
>
> **2). Regarding the concern of center point order.** Yes, the order of the center points is preserved throughout the diffusion process. In Section 3.1, we describe how the input point cloud is divided into g patches using FPS to select patch centers, followed by KNN to form patches. These centers are assigned a fixed order based on the FPS algorithm. During the forward diffusion process, Gaussian noise is added to the center coordinates independently for each center, preserving their original indexing. Similarly, in the reverse diffusion process, the encoder predicts clean centers while maintaining their correspondence to the original patch indices. The fixed order ensures that the predicted centers $\hat{C}_i^v$ and $\hat{C}_i^m$ align directly with their ground-truth counterparts $C_i^v$ and $C_i^m$, enabling the use of MSE for accurate loss computation.
>
> **\<Q4\>** We agree that their inclusion improves the completeness of our evaluation and better contextualizes our contributions. Accordingly, we have updated the comparisons in Section 4.1 to incorporate both methods.
>
> **\<Q5\>** To validate this, we conducted experiments on the SUN RGB-D dataset following the official PiMAE pipeline. We incorporate our proposed Point-MaDi into the PiMAE framework (cetner diffusion & patch diffusion process) The model is pre-trained for 400 epochs with a batch size of 16. Due to GPU resource constraints and limited rebuttal time, the fine-tuning was performed with 300 epochs.
>
> Our fine-tuned model slightly surpasses the reproduced PiMAE under the same conditions, validating the generalizability of our approach to scene-level, multi-modal pre-training. We will release our code to support further validation and reproducibility.
>
> | Method         | Pre-trained   | Epoch | AP25 | AP50 |
> |----------------|----------------|--------|------|------|
> | PiMAE| SUN RGB-D     | 1080| 59.4| 33.2 |
> | PiMAE-reproduce | SUN RGB-D     | 300    | 58.6| 31.0 |
> | PiMAE-ours      | SUN RGB-D     | 300    | 58.7| 31.2 |

---

> > ### Comment · Reviewer_wMGd · 2025-08-04
> >
> > Thanks the authors for their detailed rebuttal. The rebuttal has indeed resolved most of my concerns. I comment only to make sure I have thoroughly understood the authors' design in preventing geometric information leakage.
> >
> > From your rebuttal, the geometric information leakage is prevented via:
> > - the **pretext** denoising task to predict masked centers based on visible centers in the Encoder (where each patch feature is separately encoded with a mini Pointnet)
> > - the stop gradient design
> >
> > Therefore, if only taking the goal of preventing geometric information leakage into account, is it correct that given the predicted centers from the encoder, the decoder part can be replaced by any reconstruction method, as long as the same gradient stop design is implemented?

---

> > > ### Author Response · Authors · 2025-08-04
> > >
> > > We thank the reviewer for the insightful comment and are pleased that our rebuttal has addressed most of the concerns. We appreciate the opportunity to further clarify the design principles of our framework.
> > >
> > > Yes, the decoder can in principle be replaced by any reconstruction method, as long as the same conditions are maintained: positional embeddings are derived from predicted centers with stop-gradient applied, and no ground-truth masked patch information is accessible.
> > >
> > > Indeed, the core mechanism preventing geometric information leakage in the encoder-decoder framework lies not in the decoder architecture, but in principled safeguards across both forward and backward passes. In the forward pass, relying solely on data augmentation or masking is insufficient, as such operations do not fundamentally alter the underlying geometric information (spatial proximity and structural coherence remain), enabling the model to implicitly recover or copy structure through identity-like mappings. To prevent this, positional information must be actively predicted under meaningful perturbation rather than directly used. In the backward pass, even if predicted positions are used for conditioning, a stop-gradient or equivalent isolation must be applied to prevent gradients from the reconstruction loss in the decoder from influencing the encoder. This breaks any backpropagation path that could allow the encoder to adapt its predictions based on downstream needs. Only when both conditions are satisfied **(robust prediction without copying in the forward pass, and gradient isolation in the backward pass)** can true geometric independence be achieved.
> > >
> > > Once again, we sincerely appreciate the time and effort the reviewer have dedicated to reviewing our paper. If you have any further questions or suggestions, please do not hesitate to let us know.

---

> > > > ### Comment · Reviewer_wMGd · 2025-08-05
> > > >
> > > > Totally understand. The authors have addressed the primary concern I raised. The issue of information leakage is important to the community, and I’m glad to see this paper explicitly tackling it. I will raise my rating to 5. Please refine the writing in the final version to clearly highlight how information leakage is prevented and how this improves pre-training effectiveness.

---

> > > > > ### Author Response · Authors · 2025-08-05
> > > > >
> > > > > Thank you very much for your thoughtful and kind comments. We sincerely appreciate your time and the positive recognition of our work. As suggested, we will carefully refine the writing in the final version to explicitly clarify (1) the mechanisms for preventing information leakage and (2) how this enhancement contributes to pre-training effectiveness.
> > > > >
> > > > > Thank you again for your supportive and constructive review !

---

### Official Review · Reviewer_EQeP · 2025-06-29

**Clarity:** 3
**Significance:** 3
**Originality:** 3
**Rating:** 4
**Confidence:** 4

**Summary:**

This work proposes a novel point cloud pre-training algorithm, Point-MaDi, which integrates masked autoencoding and diffusion models for representation learning. Specifically, the method introduces two diffusion mechanisms: a center diffusion mechanism and a patch diffusion mechanism, where the latter is primarily used to prevent geometric information leakage. Extensive experiments conducted on diverse datasets demonstrate the effectiveness of the proposed approach, surpassing previous state-of-the-art methods.

**Questions:**

1. In Figure 1, the comparison with contrastive learning and masked autoencoder paradigms fails to clearly highlight the unique contribution of the proposed method. To improve this, the authors are encouraged to include comparisons with more relevant baselines such as DiffPMAE and PointDiff, and clarify the distinct advantages of Point-MaDi.

2. The authors should provide deeper qualitative and quantitative analysis to compare the effectiveness of using noised center point embeddings versus original (non-noised) center point embeddings.

3. Figure 2 is not referenced in the main text. Additionally, since the figure contains multiple subplots, it is recommended to add (a), (b), (c) labels to improve clarity. Notations such as E and PE in Eq. (7) should also be clearly indicated within the figure for better alignment with the text.

4. The reviewer suggests that Section 3 should include an overview of the entire pipeline, along with a clear explanation of how Point-MaDi is related to or differs from PointDiff.

5. In Eq. (3), it appears that the center points are already perturbed with noise, but the authors do not provide sufficient explanation or corresponding notations to clarify this process.

6. Time embedding is not discussed within the center diffusion process. The authors should provide more explanation and ideally support it with ablation experiments.

7. Are center diffusion and patch diffusion sampled simultaneously? If they follow different sampling schedules (i.e., two different time steps t), what is the effect? The authors are encouraged to add comparative experiments to explore this question.

8. It remains unclear what specific types of information are learned by the encoder and decoder through the center and patch diffusion mechanisms, respectively. Providing visualizations of the learned representations would help the reader better appreciate the contributions of this work.

**Ethical Concerns:**

["NO or VERY MINOR ethics concerns only"]

**Final Justification:**

I appreciate the authors' rebuttal, which has addressed most of my concerns. At this stage, I intend to maintain my original score.

**Limitations:**

yes

**Quality:**

3

**Strengths And Weaknesses:**

Strengths:

1.A simple yet effective design that facilitates engineering extensibility.

2.The paper is clearly written and easy to follow.

Weaknesses:

1.Insufficient qualitative comparisons with existing approaches.

2.Limited interpretability and ablation analysis for the individual components.

---

> ### Author Rebuttal · Authors · 2025-07-30
>
> We are grateful to the reviewer for the thoughtful and encouraging comments. The provided suggestions have led us to refine our presentation and deepen the analysis of our method’s contributions. Below, we respond to each concern with detailed clarifications and additional experiments.
>
> **\<Q1\> Lack of clear comparison with relevant diffusion-based baselines.** In the revised version, we will update Figure 1 by replacing the schematic representations of contrastive learning and MAE with two more pertinent methods: DiffPMAE and PointDiff. We will also briefly describe the characteristics of these methods.
>
> **Revised version:** “Figure 1 illustrates how Point-MaDi contrasts with existing pretext diffusion tasks: while DiffPMAE applies diffusion to masked tokens only within a masked autoencoding framework, PointDiff treats pre-training as conditional generation over full point clouds via diffusion. In contrast, our Point-MaDi introduces a dual diffusion framework that denoises both visible centers and masked patches through aligned encoder-decoder objectives, enabling robust and geometry-aware representation learning without relying on positional priors.”
>
> **\<Q2\> Concern about the effectiveness of using noised centers versus clean ones.** We conducted a comprehensive ablation study focusing on the components of our dual-diffusion framework as below. To ensure a fair comparison, we maintained the core Point-MaDi framework. The baseline configuration uses clean patch centers for both visible and masked patches in the encoder and employs learnable mask tokens in the decoder. The key variable is the activation or deactivation of the diffusion processes. The analysis reveals several key points: firstly, the baseline achieves a performance of 93.97, 92.60, 88.83 on OBJ_BG, OBJ_ONLY, PB_T50_RS. When only one set of centers is noised while the other remains clean, the performance compared to the baseline is either slightly degraded or shows mixed results, suggesting that providing clean positional information for only a subset of patches creates an inconsistent learning signal for the encoder; secondly, when both visible and masked centers are noised in the encoder, performance improves to 94.32, 92.94, 89.17, demonstrating that our full center diffusion mechanism, which removes all ground-truth positional shortcuts and forces the encoder to predict all clean centers from noisy inputs, is more effective than partial noising or relying on clean centers. Furthermore, this benefit is amplified when combined with the patch diffusion task in the decoder: the configuration with full center diffusion and patch diffusion enabled achieves our best overall performance, significantly outperforming the scenario where center diffusion is disabled but patch diffusion is active. Additionally, the analysis highlights the positive impact of incorporating time embeddings for patch diffusion processes.
>
> | Center (Vis) | Center (Mask) | Patch | Time Embedding | OBJ\_BG | OBJ\_ONLY | PB\_T50\_RS |
> | ------------ | ------------- | ----- | -------------- | ------- | --------- | ----------- |
> | ✗            | ✗             | ✗     | ✗              | 93.97   | 92.60     | 88.83       |
> | ✓            | ✗             | ✗     | ✗              | 93.63   | 92.43     | 88.13       |
> | ✗            | ✓             | ✗     | ✗              | 94.32   | 92.08     | 88.79       |
> | ✓            | ✓             | ✗     | ✗              | 94.32   | 92.94     | 89.17       |
> | ✗            | ✗             | ✓     | ✓              | 94.66   | 93.11     | 89.17       |
> | ✓            | ✓             | ✓     | ✗              | 94.49   | 92.43     | 88.83       |
> | ✓            | ✓             | ✓     | ✓              | 95.53   | 93.46     | 89.52       |
>
> **\<Q3\> Figure 2 is not referenced and lacks proper subfigure labels and alignment with equations.** We sincerely thank the reviewer's constructive suggestions, which are indeed very helpful for improving the clarity and overall presentation of our paper. We are committed to implementing the suggested changes: Specifically, we will explicitly reference Figure 2 in the main text, particularly within the newly added subsection that introduces the overall Point-MaDi pipeline. We will also revise Figure 2 to include (a), (b), and (c) labels for its subplots to enhance clarity. In addition, we will refine the figure’s legend and annotations to clearly indicate key notations such as E and PE, ensuring consistency with the corresponding text. Due to the constraints of the rebuttal process, we are unable to include the revised figure at this stage. However, we have already prepared the necessary updates and will incorporate them into the revised version of the paper to improve its clarity and completeness.
>
> **\<Q4\> Missing overview of the full pipeline.** We have revised the corresponding section to include a concise pipeline description and highlight the key distinctions from PointDiff.
>
> **Revised version:** “3.1 Point-MaDi Framework
> Point-MaDi comprises two diffusion-driven modules tailored for structural and geometric representation learning. As illustrated in Fig.1, given a clean point cloud, the encoder partitions it into $g$ patches and applies a diffusion model to progressively noise the centers of both visible and masked patches. The model then learns to denoise these centers. Conditioned on the visible tokens and predicted centers, the decoder reconstructs the masked patches under Chamfer Distance supervision. Unlike PointDiff, which performs point-wise generation via a multi-module architecture, Point-MaDi unifies center and patch diffusion within a masked autoencoding framework, enabling data-driven pretraining without reliance on ground-truth positional embeddings.”
>
> **\<Q5\> Insufficient explanation of center perturbation process.** We apologize for the unclear presentation of the notation in the initial submission. To address this confusion, in the revised manuscript, we will explicitly clarify that the center coordinates ${C^v}$ and ${C^m}$ used in the positional embedding calculation.
>
> **Revised version:** “where $d$ is the hidden dimension of the network. To incorporate spatial information, we compute positional embeddings for both visible and masked patches. The original centers ${C^v}$ and ${C^m}$ at  are progressively noised into $C_t^v$, $C_t^m$ in the subsequent center diffusion process, which are then used as inputs to the MLP for computing positional embeddings:”
>
> **\<Q6\> Missing discussion and ablation of time embeddings in the center diffusion process.** In our framework, we intentionally omit time-step embeddings in the transformer encoder during the center diffusion process. The motivation behind this decision is to align the encoder's architecture with downstream tasks, where no diffusion steps or time conditioning are present. By avoiding the introduction of time embeddings during pretraining, we ensure that the encoder learns time-agnostic, task-agnostic features that generalize better across downstream domains. To validate this design decision, we conducted an ablation study comparing two variants: 1) With time embedding added to the encoder; 2) Without time embedding, as used in our default implementation. The results on the ScanObjectNN benchmark are summarized below. Incorporating time embeddings led to consistent performance degradation across all three variants. We hypothesize that the time embeddings may introduce unnecessary conditioning noise or reduce the encoder’s ability to generalize to downstream inputs, which are always clean and have no associated time-step semantics.
>
> | Time Embedding | OBJ\_BG | OBJ\_ONLY | PB\_T50\_RS |
> | -------------- | ------- | --------- | ----------- |
> | ✗              | 94.49   | 92.59     | 88.83       |
> | ✓              | 95.53   | 93.46     | 89.52       |
>
> **\<Q7\>  Lack of clarification on whether center and patch diffusion follow joint or separate schedules.** In our default setting, center diffusion and patch diffusion are sampled simultaneously using the same number of time steps ${t_c} = {t_p} = 2000$ with a shared linear variance schedule (${\beta_t}$ from 0.0001 to 0.02).  We conducted ablation experiments by varying ${t_c}$ and ${t_p}$ independently. We observe that desynchronizing the diffusion schedules leads to performance degradation in all cases. Interestingly, the model is more sensitive to changes in the decoder-side patch diffusion ${t_p}$ compared to variations in the encoder-side center diffusion ${t_c}$. When ${t_p}$ is reduced while keeping ${t_c}$ fixed (e.g., ${t_p} = 200$), the performance drops more significantly, likely due to insufficient corruption during training, which weakens the decoder’s ability to reconstruct complex local geometry. In contrast, reducing ${t_c}$ while keeping ${t_p}$ fixed leads to a smaller, though still noticeable, performance drop.
>
> | \$t\_c\$ | \$t\_p\$ | OBJ\_BG | OBJ\_ONLY | PB\_T50\_RS |
> | -------- | -------- | ------- | --------- | ----------- |
> | 200      | 2000     | 94.15   | 92.60     | 88.90       |
> | 1000     | 2000     | 94.15   | 92.60     | 88.72       |
> | 2000     | 2000     | 95.53   | 93.46     | 89.52       |
> | 2000     | 1000     | 93.80   | 92.94     | 88.97       |
> | 2000     | 200      | 94.66   | 91.91     | 88.72       |
>
> **\<Q8\> Lack of analysis or visualization of encoder and decoder representations.** The encoder, through the center diffusion task, is encouraged to capture global geometric structure and contextual dependencies, the decoder, via the patch diffusion task, focuses on reconstructing fine-grained local geometry, learning to recover detailed point distributions within each patch.To further enhance interpretability, we will include additional t-SNE plots of intermediate representations and segmentation-based visualizations in the final version.

---

> > ### Author Response · Authors · 2025-08-06
> >
> > Dear Reviewer,
> >
> > I hope this message finds you well. We truly appreciate your valuable feedback and the time you've taken to engage with our work. We certainly don't mean to disturb you, but since the discussion period is approaching its deadline, we just wanted to kindly check in to see if you had any remaining concerns regarding our rebuttal. We are more than happy to provide further clarification or address any additional questions you may have.
> >
> > Thank you again for your thoughtful comments and for contributing to the improvement of our work.

---

> > ### Comment · Reviewer_EQeP · 2025-08-07
> >
> > I appreciate the authors' rebuttal, which has addressed most of my concerns. At this stage, I intend to maintain my original score.

---

> > > ### Author Response · Authors · 2025-08-08
> > >
> > > We are grateful for the reviewer’s encouraging response and for acknowledging our revisions. The constructive comments and engagement have been instrumental in refining our work, and we sincerely appreciate the thoughtful evaluation.

---

### Official Review · Reviewer_tX9U · 2025-07-21

**Clarity:** 2
**Significance:** 2
**Originality:** 2
**Rating:** 3
**Confidence:** 4

**Summary:**

This paper, "Point-MaDi: Masked Autoencoding with Diffusion for Point Cloud Pre-training," proposes a novel self-supervised pretraining framework for point clouds called Point-MaDi. Its core innovation lies in integrating a dual-diffusion process into the Masked AutoEncoder (MAE) architecture. This design eliminates the reliance on geometric priors—such as explicit center position embeddings—found in traditional methods, while enhancing the model’s ability to learn both local geometric details and global structural representations directly from the data.

**Questions:**

It seems that the center diffusion module is not utilized during downstream inference, and I couldn't find a clear description of this in the paper.

I recommend that the authors explicitly clarify how the pretrained model is used for downstream tasks. If I have overlooked this detail, please kindly point out where it is mentioned in the paper.

**Ethical Concerns:**

["NO or VERY MINOR ethics concerns only"]

**Final Justification:**

I still stand by my opinion that patch positional encoding represents only very coarse-level information of the point cloud, and performing diffusion on top of it is unnecessary. Moreover, if you downsample the point cloud, those points may become so coarse that they resemble noise and may not follow any specific distribution, making diffusion on them even less reasonable.
In short, I consider this to be a borderline-below paper. I don’t understand why other reviewers have given scores that exceed expectations. I will keep my original score, leaving the final decision to the AC.

**Limitations:**

I believe the theoretical motivation behind this paper's contribution is not particularly strong. Although the method may show some empirical improvements, these could be attributed to factors such as model design or the amount of training data used. Overall, I find the authors’ justification for the methodological motivation unconvincing.

**Quality:**

2

**Strengths And Weaknesses:**

The main contribution of this paper is the introduction of a patch position diffusion process prior to patch-level MAE modeling on point clouds. While this design sounds more reasonable, it seems somewhat similar to a form of data augmentation on patch positions. However, I am not entirely convinced by the authors’ claim that this design helps the model better understand structural information.

This is because patch positions only represent very coarse-grained information about the point cloud. How different can the predicted patch centers obtained from the center diffusion be compared to the ground-truth patch center positions? If the difference is minimal, then this might be equivalent to simply adding small perturbations to the ground-truth patch centers. On the other hand, if the predicted centers deviate significantly, they might no longer be suitable as positional embeddings (PE) for the patch diffusion decoder.

Even if the model does learn certain geometric structures, such knowledge is acquired within the center diffusion process itself, not necessarily transferred downstream.

---

> ### Author Rebuttal · Authors · 2025-07-29
>
> We sincerely thank the reviewer for the constructive feedback and careful evaluation. The concerns raised regarding the center diffusion mechanism and its impact on downstream transfer have prompted us to revisit and clarify several key aspects of our design. We address each point below with detailed explanations and supporting results.
>
> **\<W1\> Concern about the effectiveness and necessity of the center diffusion process.** To address this point, we organize our response into three aspects:
>
> **1). Regarding the distinction from data augmentation.** We fully agree that patch center coordinates represent coarse-grained spatial information. However, the core innovation of Point-MaDi's center diffusion lies not in perturbing these coordinates (as typical data augmentation does), but in removing the reliance on ground-truth positional embeddings and forcing the encoder to infer global structure from partial, noisy observations.
>
> Standard data augmentation (e.g., adding noise to coordinates) aims to improve model robustness by creating slightly varied inputs while preserving the original semantics. The perturbations are usually mild and do not fundamentally alter the data's structure. Moreover, data augmentation does not require learning; it's non-parametric and uncontrolled. In contrast, our center diffusion process, over 2000 steps, progressively degrades the original patch center coordinates $C_{0}^{v}$, $C_{0}^{m}$ into near-Gaussian noise $C_{t}^{v}$, $C_{t}^{m}$. This drastic corruption means the encoder cannot rely on any form of original positional information. It is then tasked with predicting the clean centers solely based on the visual content and contextual relationships derived from the visible and masked patches. This constitutes a "location-free" learning paradigm, compelling the model to develop an internal understanding of spatial layout, rather than merely being robust to minor input variations.
>
> **2). Regarding learning structural information.** Building on the above, once ground-truth center positions are no longer accessible due to the heavy corruption in diffusion, the encoder must leverage self-attention among visible patches and cross-attention from masked to visible patches to reason about the likely positions of all patches. Predicting centers from high noise is not a trivial interpolation but a complex inference based on learned geometric priors and relationships. The predicted centers, even if not pixel-perfect, represent the model's synthesized understanding of the global structure.
>
> **3). Regarding difference between predicted and ground-truth center.** We acknowledge the predicted centers $\hat{C}$ may differ from the ground-truth $C$. However, our experiments (e.g., superior downstream performance, visualizations in Appendix Fig 6-8 showing alignment) demonstrate that these predictions are meaningful and capture essential structural aspects. Even with deviations, they provide far more contextual guidance than random or grossly incorrect positions.
>
> To further validate this, we report per-category Chamfer Distance ($×10^2$) on ShapeNe test. Lower errors in rigid categories (e.g., guitar, rifle, airplane) indicate that the predicted centers remain close to the ground-truth and preserve their positional relevance.
>
> | Category     | microphone | earphone | guitar | laptop | table | chair | airplane | car  | rifle | motorbike | pistol |
> | :----------- | :--------: | :------: | :----: | :----: | :---: | :---: | :------: | :--: | :---: | :-------: | :----: |
> | CD ↓         |    2.60    |   4.78   |  0.73  |  2.83  | 4.07  | 2.67  |   1.43   | 2.29 | 1.04  |   2.68    | 0.02 |
>
> **\<W2\> Concern about whether the learned structural knowledge is transferred to downstream tasks.** To clarify how the structural knowledge learned during pretraining benefits downstream tasks, we elaborate on the encoder design and provide empirical evidence as follows:
>
> **1). Unified encoder design enables joint learning of geometry and representations.** The encoder in our framework serves a dual role during pretraining: **it performs the center denoising task by predicting clean patch centers $C^v$, $C^m$, and it simultaneously extracts feature representations $T^v$ for masked patch reconstruction. This shared transformer encoder is the core of both pretext objectives.**
>
> Learning to accurately recover patch center positions from their noisy counterparts requires the encoder to capture global spatial context and inter-patch geometric relationships, rather than relying on absolute positions or local clues. As such, the learned structure is not isolated to a separate branch but becomes an intrinsic part of the encoder’s output representations.
>
> **2). Encoder-only downstream protocol ensures full reliance on learned representations.** Once pretraining is complete, we discard the decoder entirely, and only the pretrained encoder is retained for downstream tasks such as classification, segmentation, and detection. Therefore, any performance in downstream tasks relies entirely on the learned encoder features, not on decoder-specific logic or intermediate reconstruction tricks.
>
> **3). Empirical and ablation evidence confirms transferability of structural knowledge.** We present two lines of evidence to validate that the geometric knowledge acquired during center diffusion is effectively transferred to downstream tasks:
>
> - **Ablation study on positional shortcuts**:
>    We conduct a control experiment in which the encoder is provided with clean patch centers during pretraining instead of learning to denoise them. This simulates a positional shortcut akin to standard positional embeddings. We find that this variant performs worse across multiple downstream tasks. In contrast, our diffusion-based center corruption and recovery removes the positional prior and forces the model to internalize structural reasoning, leading to stronger generalization. This ablation directly supports our motivation and validates the necessity of the center diffusion design.
>
> - **Performance on structure-sensitive benchmarks**:
>    Point-MaDi outperforms Point-MAE across a wide range of downstream tasks. For instance, we observe a +5.51% gain on ScanObjectNN (OBJ-BG), +0.6% mIoU on ShapeNetPart, and +0.9% AP@50 on ScanNet 3D detection — all tasks that require strong structural awareness. These consistent gains indicate that the encoder’s learned features capture transferrable geometric information beyond appearance or local patterns.
>
> **\<Q1\>  Concern about whether the center diffusion module is used during downstream inference.**
>
> Indeed, the center diffusion module is only employed during the pretraining phase as part of the self-supervised pretext task. During downstream inference and fine-tuning, we do not include the center diffusion process, nor do we perform any additional denoising steps. Instead, we follow the standard protocol adopted by prior masked autoencoder works (e.g., Point-MAE, Point-BERT) and retain only the pretrained encoder to extract representations from clean, uncorrupted input point clouds.
>
> **\<Q2\> Suggestion to clarify how the pretrained model is used for downstream tasks.**
>
> We sincerely apologize for the confusion and appreciate you raising this important point, and acknowledge that the paper could have been clearer on this matter. Following the standard protocol established by prior work (e.g., Point-MAE, Point-BERT), after the pretraining phase is complete: 1. The decoder (responsible for patch diffusion reconstruction) is discarded. 2. The encoder, which has learned rich representations through the dual-diffusion pretext task, is retained. 3. The pretrained encoder is then directly fine-tuned (or used with a task-specific head for linear evaluation) for various downstream tasks (classification, segmentation, detection).
>
> We thank the reviewer once again for pointing out this potential source of confusion. We will explicitly clarify this standard procedure in the revised version of the manuscript, likely in the dedicated section on implementation details for downstream tasks.

---

> > ### Comment · Reviewer_tX9U · 2025-07-31
> >
> > You still haven't quite gotten my point. If the structural information truly leaks from the positional embeddings as you claim, then simply applying data augmentation directly on the positional embeddings should already lead to performance improvement. If there is no improvement, then your claim doesn't hold.
> >
> > Your method may be better, but I'm not at all convinced of that. At the very least, you need to demonstrate that your approach outperforms straightforward augmentation on positional embeddings, rather than relying on weak verbal explanations.

---

> > > ### Author Response · Authors · 2025-08-06
> > >
> > > Dear Reviewer,
> > >
> > > We would be grateful if you could let us know whether our rebuttal has addressed your concerns. Please don’t hesitate to reach out if you have any remaining questions or suggestions! We sincerely appreciate the time and effort you have dedicated to reviewing our paper.

---

> ### Author Response · Authors · 2025-08-02
>
> We thank the reviewer for the continued engagement with our work and for raising this critical question. We conducted a series of additional experiments comparing our center diffusion process against three standard forms of data augmentation applied directly to patch centers.
>
> We implemented and evaluated three representative types of augmentation applied consistently to both visible and masked patch centers: scale & jitter, scale & translation & rotation, translation, our proposed center diffusion method, and a clean baseline without any augmentation. All five variants were pretrained under identical settings, and we report both reconstruction losses and downstream classification performance on three standard benchmarks.
>
> | Perturbation Strategy    |  ${{\cal L}_{{\rm{center}}}}( \times {10^3})$ | OBJ_BG         | OBJ_ONLY       | PB_T50_RS      |
> |--------------------------|--------------------|----------------|----------------|----------------|
> | No                       | 0.0                | 93.97          | 92.60          | 88.83          |
> | Scale & Jitter           | 0.3                | 94.49 (+0.52)  | 92.25 (−0.35)  | 89.24 (+0.41)  |
> | Scale & Translate & Rotate | 5.2              | 93.97 (+0.00)  | 92.43 (−0.17)  | 88.97 (+0.14)  |
> | Translate                | 5.3                | 94.49 (+0.52)  | 92.25 (−0.35)  | 89.27 (+0.44)  |
> | **Center Diffusion**         | **29.0**           | **95.53 (+1.56)** | **93.46 (+0.86)** | **89.52 (+0.69)** |
>
> The experimental outcomes provide several important insights. First, standard augmentations do yield mild and sometimes unstable gains in downstream performance compared to the clean baseline. For instance, Scale & Jitter achieves +0.52% on OBJ_BG and +0.41% on PB_T50_RS, indicating that perturbing patch centers may enhance robustness to small spatial variations in our framework. However, these gains are relatively limited, and sometimes inconsistent across different datasets (-0.35% on OBJ_ONLY). Moreover, the center loss remains near zero, suggesting that the model still retains access to reliable positional cues. As a result, the reconstruction task remains a relatively low-effort matching problem, where shortcut solutions may still suffice. In contrast, center diffusion introduces a significantly more challenging task, as evidenced by the much higher center loss (29.0). This degradation prevents reliance on original spatial priors and forces the model to reconstruct the scene structure from noisy, diffusion-corrupted inputs. Despite this difficulty, center diffusion achieves the best and more consistent performance across all benchmarks, indicating that the model has developed a stronger and more transferable understanding of spatial layout and geometry.
>
> **These findings empirically support our initial claim: conventional augmentation does not fundamentally alter the nature of the task, whereas center diffusion transforms it from a shortcut-prone reconstruction problem into a structure-aware inference problem. We will release all code and training logs to ensure full reproducibility.**

---

### Note · Authors · 2025-08-13

We sincerely thank the AC and all reviewers for your dedication, and constructive engagement throughout the review process. Your questions have significantly helped us clarify and strengthen the presentation of our work.

Point-MaDi proposes a dual-diffusion framework for point cloud pre-training that eliminates reliance on explicit geometric priors. We introduce center diffusion, where patch centers are progressively corrupted into noise and must be predicted by the encoder, enabling the model to learn global structure from contextual cues alone. Simultaneously, masked patches are reconstructed from noise in the decoder, conditioned on predicted centers and visible features. This unified design fosters truly data-driven learning of both global semantics and local geometry, leading to strong generalization across classification, segmentation, and detection.

A major concern from Reviewer **tX9U** is whether center diffusion meaningfully differs from data augmentation. We clarify it is not augmentation: over 2000 steps, centers are corrupted into near-Gaussian noise, making original positions inaccessible. Ablation shows standard augmentations yield minor, inconsistent gains, while center diffusion improves performance by +1.56% on OBJ_BG with larger gains on structure-sensitive tasks. This supports its role in structural reasoning.

Reviewer **EQeP** raised important points on component analysis. We confirm center and patch diffusion are synchronized; desynchronization degrades performance. Time embeddings are omitted in the encoder to ensure time-agnostic features. We will update Figure 1 to compare with DiffPMAE and PointDiff, and enhance figure clarity with subfigure labels and notation alignment.

Reviewer **wMGd** questioned whether geometric leakage is truly avoided, especially given the use of predicted centers in the decoder. We emphasize that no ground-truth positions are used. Predicted centers are inferred by the encoder and passed with SG, preventing backpropagation leakage. This design avoids explicit priors while providing necessary spatial context for reconstruction.

Reviewer **LUo6** highlighted efficiency and visualization. Our method adds only marginal cost over Point-MAE  while outperforming it significantly. We will include per-category center prediction analysis and segmentation visualizations in the final version.

Once again, we sincerely thank the reviewers’ suggestions and believe our responses have addressed the core concerns.

---

### Decision · Program_Chairs · 2025-09-17

**Decision:**

Accept (poster)

**Comment:**

This paper proposes a diffusion-based point cloud pre-training algorithm, Point-MaDi, Specifically, the method introduces two diffusion mechanisms: a center diffusion mechanism and a patch diffusion mechanism, where the latter is primarily used to prevent geometric information leakage. Extensive experiments conducted on diverse datasets demonstrate the effectiveness of the proposed approach, surpassing previous state-of-the-art methods.

Three reviewers are positive about the paper before/after the rebuttal. One reviewer remains skeptical about the paper, claiming patch-based encoding in the paper is sub-optimal. The authors did provide experimental results that justify the empirical performance gains. The AC recommends acceptance due to the majority vote among the reviewers. It is recommended that the authors provide more intuitions about the proposed patch encoding scheme in the final version.